# Scaling Open-Vocabulary Object Detection

**Matthias Minderer**      **Alexey Gritsenko**      **Neil Houlsby**

Google DeepMind
{mjlm, agritsenko, neilhoulsby}@google.com

## Abstract

Open-vocabulary object detection has benefited greatly from pretrained vision-language models, but is still limited by the amount of available detection training data. While detection training data can be expanded by using Web image-text pairs as weak supervision, this has not been done at scales comparable to image-level pretraining. Here, we scale up detection data with self-training, which uses an existing detector to generate pseudo-box annotations on image-text pairs. Major challenges in scaling self-training are the choice of label space, pseudo-annotation filtering, and training efficiency. We present the OWLv2 model and OWL-ST self-training recipe, which address these challenges. OWLv2 surpasses the performance of previous state-of-the-art open-vocabulary detectors already at comparable training scales (≈10M examples). However, with OWL-ST, we can scale to over 1B examples, yielding further large improvement: With a ViT-L/14 architecture, OWL-ST improves AP on LVIS rare classes, *for which the model has seen no human box annotations*, from 31.2% to 44.6% (43% relative improvement). OWL-ST unlocks Web-scale training for open-world localization, similar to what has been seen for image classification and language modelling. Code and checkpoints are available on GitHub.[1]

## 1 Introduction

Object detection is a core computer vision task with many real-world applications. Consequently, there is great interest in improving detection models, especially in the open-vocabulary domain. For image-level tasks, large improvements have been achieved through contrastive pretraining of vision-language models, which is massively scalable because it can use naturally abundant weak supervision in the form of image-text pairs from the Web [30, 12, 29]. Since no such natural supervision data exists for localization tasks, open-vocabulary detection models typically build on pretrained image-level encoders [9, 19, 26, 46, 22, 1, 39, 47]. However, due to the scarcity of detection data and the fragility of pretrained representations, detection-training stages of these models have typically had to be relatively brief, which limits final detection performance and scaling potential.

The scarcity of detection data can be addressed with *self-training*. In self-training, an existing detector is used to predict bounding boxes on unlabeled images to generate data for training better detectors [31, 48, 35]. By combining open-vocabulary detectors with Web image-text data, such pseudo-labeling can produce practically unlimited amounts of open-vocabulary detection training data that leverages the image-associated text for semantic supervision. While several works have applied various forms of self-training to open-vocabulary object detection [46, 1, 47, 39, 38], they have done so at relatively small scales, comparable to the size of human-annotated detection datasets and much smaller than the datasets used for image-level training.

---

[1] https://github.com/google-research/scenic/tree/main/scenic/projects/owl_vit

37th Conference on Neural Information Processing Systems (NeurIPS 2023).

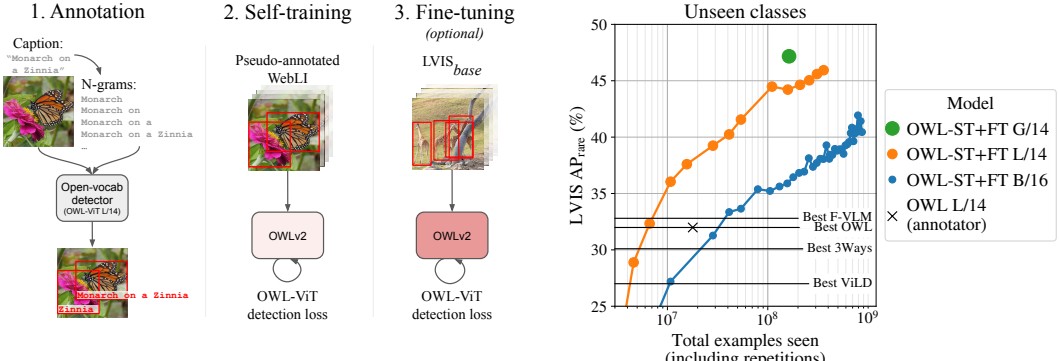

Figure 1: Overview of our method. **Left:** Our method has three steps: (1) Generate pseudo-box annotations on WebLI with OWL-ViT L/14, queried with caption N-grams. (2) Train new models on pseudo-annotations. (3) Optionally, fine-tune on human annotations. **Right:** Zero-shot detection performance on $LVIS_{rare}$ after fine-tuning on $LVIS_{base}$. Neither the annotator nor our models have seen any human-generated box annotations for $LVIS_{rare}$ classes. Our self-training approach improves over other methods even at moderate amounts of training (e.g. the OWL-L/14 model we use as annotator; black ×), and continues to improve as training is scaled up. Horizontal black lines indicate previous state-of-the-art open-vocabulary detectors which did not see $LVIS_{rare}$ classes during training.

To scale detection self-training further, we take guidance from image-level methods, where the principle has been to leverage weak supervision in the largest possible amount [30, 12, 29, 42]. We identify three key ingredients for optimizing the use of weak supervision for detection: choice of label space, filtering of pseudo-annotations, and training efficiency. Prior methods have typically used human-curated label spaces or complex concept mining [47, 39, 38, 46] and strict filtering, keeping just the single largest [47] or highest-scoring [1] pseudo-box for each image. In contrast, we argue that we should "let the data do the work" and therefore apply little processing and filtering. We propose to simply use all possible N-grams of the image-associated text as detection prompts for that image, and apply only weak confidence filtering to the resulting pseudo-labels.

We apply this self-training recipe to the OWL-ViT detection architecture [26] and call it OWL-ST. To increase the number of examples seen for a given amount compute, we also introduce OWLv2, an optimized architecture with improved training efficiency. Combining the OWL-ST recipe with the OWLv2 architecture surpasses prior state-of-the-art methods already at moderate amounts of self-training, comparable to training amounts of previous methods (Figure 1). Scaling self-training to billions of examples yields further large improvements. For example, our ViT-L/14-based model, trained on 2.3B image-text pairs and fine-tuned on $LVIS_{base}$, achieves 44.6% zero-shot LVIS $mAP_{rare}$, which is a 36% relative improvement over the prior state of the art (32.8% $mAP_{rare}$ for F-VLM R50x64 [19]). Our largest model, ViT-G/14, reaches 47.2% $mAP_{rare}$.

We also evaluate our models on a suite of "in the wild" datasets [21] and study the trade-off between fine-tuned and open-vocabulary performance. We find that strong in- and out-of-distribution performance is possible with weight ensembling [37]. Finally, our analysis of the scaling behavior of OWL-ST suggests that self-training has further potential for leveraging abundantly available weak supervision for open-vocabulary object detection.

## 2 Related Work

### 2.1 Scaling Vision Models

Vision models have recently seen large advances in model and training scale, leading to improved performance on many image-level tasks. On the architecture side, Vision Transformers have been shown to scale more efficiently than prior architectures [17]. Task performance improves predictably as training data and compute are increased [42], with recent work showing continued improvements for models with up to 22 billion parameters [6]. We apply these findings to object detection.

On the data side, contrastive pretraining of vision-language models (VLMs) [30] has unlocked the use of abundantly available image-text pairs from the Web as weak supervision, with improved results if more data is used [12, 28]. VLMs, which embed images and text into a shared space, also enable open-vocabulary applications where prior models were limited to fixed label spaces. Here, we use pretrained CLIP [30] and SigLIP [43] encoders as backbones for our detector.

## 2.2 Open-Vocabulary Object Detection

Much recent work aims to transfer the open-vocabulary capabilities of VLMs to localization tasks such as object detection. A first wave of VLM-based object detection methods either distilled VLM-predictions for cropped image regions (e.g. **ViLD** [9]), or added detection heads directly to frozen (**F-VLM** [19]) or fine-tuned (**OWL-VIT** [26]) VLM encoders. A challenge identified by these works is to protect the VLM from forgetting its open-vocabulary knowledge while training the detection heads on the relatively little available detection data.

## 2.3 Scaling Open-Vocabulary Detection with Weak Supervision

Given that earlier methods identified detection data as a limiting factor in open-vocabulary detection performance, more recent works focus on using weak supervision directly for detection training, rather than just during VLM pretraining. There are two main approaches:

Some methods use *self-training*, in which an existing detector is used to predict pseudo-boxes for images where image-level labels or captions, but no human box annotations, are available. Better detectors can then be trained on the pseudo-annotations. For example, **RegionCLIP** [46] generates pseudo-boxes using nouns parsed from image captions and uses those boxes for localization pre-training. **Detic** [47] predicts class-agnostic pseudo-boxes on images for which classification labels are available and associates the largest predicted box with the image label. Similar to our approach, **3Ways** [1] uses an existing open-vocabulary detector to predict pseudo-boxes on captioned images, but uses the whole caption as a prompt, instead of dividing it into multiple prompts as we do.

Other methods propose grounding losses that directly train a detector on weak supervision such as image-level labels or captions. These methods pretrain models to align class-agnostic pseudo-boxes with words from image-associated text and rely on human-generated detection data for fine-tuning. Major examples of this approach are **GLIPv1/v2** and [22, 45] and **DetCLIPv1/v2** [39, 38].

In principle, these approaches unlock Web-scale training for detection, but prior methods rarely go much beyond 10M examples and instead focus on the model architecture and training loss. Here, we keep architecture and loss simple, and focus on scaling up the training data, since this was successful for image-level models. A similar approach was recently applied with good results to class-agnostic segmentation in the **Segment Anything** work [16]. Together with our results on text-conditioned localization, this suggests that scaling up self-training is a powerful and general method for improving performance on fine-grained vision tasks.

# 3 Method

We propose a simple self-training approach with three steps: (1) Use an existing open-vocabulary detector to predict bounding boxes for a large Web image-text dataset. (2) Self-train a new detector on the pseudo-annotations. (3) Optionally, fine-tune the self-trained model briefly on human-annotated detection data (Figure 1, left). Our goal is to optimize the key components of this approach—label space, annotation filtering, and training efficiency—such that it provides strong and scalable open-vocabulary performance with few human annotations.

## 3.1 Generating Web-Scale Open-Vocabulary Object Annotations

We use the WebLI dataset [4] as the source of weak supervision for self-training. WebLI is a large dataset of images and texts available on the public Web. The dataset consists of approximately 10B images and associated alt-text strings, which can be thought of as noisy image captions. For images whose alt-text is not in English, we use an automatically generated English translation [4].

We use OWL-ViT CLIP-L/14 [26] to annotate all 10B WebLI images with bounding box pseudo-annotations. OWL-ViT is an open-vocabulary object detector. Given an image, the model first detects objects in the image in a class-agnostic way. Then, given a list of free-text queries, the model produces scores indicating the likelihood that each detected object is associated with each text query.

A crucial design choice for open-vocabulary pseudo-labeling is the annotation label space. Methods in the literature vary widely but typically fall somewhere between two extremes: (1) use a fixed, human-curated label space for all images (e.g. [47]), or (2) machine-generate per-image queries from image-associated text (e.g. [1]). We implement both and compare their performance in Section 4.3.

**Human-curated label space.** We performed one pseudo-annotation run by combining the label sets from the LVIS [10], Objects365 [33], OpenImagesV4 [20], and Visual Genome [18] datasets and removing duplicates and plural forms. In total, this label space contains 2520 common object categories, e.g. `"phone"`, `"goatee"`, `"teakettle"`, `"park"`, `"suit (clothing)"`. See Appendix A.2 for code to generate the full list. Models trained on this label space may not be considered fully *open-vocabulary* for evaluation datasets whose classes were included in the pseudo-annotation label space (e.g. LVIS), since the evaluation vocabulary is known at training time in this case. However, LVIS$_{rare}$ classes are still *unseen* for all of our models, in the sense that neither the annotator nor the self-trained models have ever seen human box annotations for LVIS$_{rare}$ classes.

**Machine-generated label space.** In a second pseudo-annotation run, we automatically generated queries from the image-associated text. Prior work using image captions as weak supervision for detection often used grammatical parsing to extract noun phrases or concepts [46, 39, 38]. These approaches may add biases that reduce the diversity of extracted queries. To keep such biases to a minimum, we use no grammatical parsing and simply extract all word N-grams up to length 10 from the text associated with a given image and use them as queries for that image. We apply minimal filtering, only removing generic terms like `image` or `png`, and queries consisting entirely of stop-words (details in Appendix A.3). Note that, since OWL-ViT uses late image-text fusion, the quality of box *localization* (as opposed to classification) is not affected by the chosen label space.

Regardless of label space, we ensemble predictions over seven prompt templates such as `"a photo of a {}"` as described in [26]. For each predicted box, we keep the query with the highest score as its pseudo-label. For each image, we keep all boxes above a score threshold. We study the choice of threshold in Section 4.4. The pseudo-annotations are used as hard labels for self-training.

## 3.2 Self-training at Scale

We now describe how we use the pseudo-annotations to train better detectors. We use a variant of the OWL-ViT architecture [26] as described below. The image and text encoders are initialized from contrastively trained image-text models (CLIP, unless noted otherwise); the detection heads are randomly initialized. All models are first trained exclusively on pseudo-annotations ("self-training"). In an optional separate step, models are fine-tuned briefly on human-annotated detection data.

Self-training proceeds similarly to detection training in [26]. In particular, we use the same losses and also augment queries with "pseudo-negatives" that are randomly sampled from the queries of other images, similar to batch negatives in [1]. Due to the size of our dataset, in contrast to [26], we use no random prompt templates and fewer image augmentations (details in Appendix A.5).

Prior work on image-level tasks shows that pretraining improves performance on downstream tasks well beyond 1 billion examples seen [44, 12, 28, 42], across model sizes. We hypothesize that similar scaling applies to detection self-training. We therefore optimize training efficiency to maximize the number of images seen for a given amount of training compute as follows.

**Token dropping.** Vision Transformers represent images as an unordered sequence of tokens. Tokens can therefore be reorganized or dropped without changing the model parameters. Various forms of token dropping or pooling have been proposed to improve efficiency [24, 32, 41, 25, 2]. Here, we drop tokens simply based on the pixel variance of the corresponding image patch. Both natural and Web images contain low-variance areas devoid of useful information, e.g. sky, single-color backgrounds, or padding. We find that the lower half of image patches by mean pixel variance can be dropped without loss in detection performance (Appendix A.6). We therefore drop 50% of patches in all of our experiments during training. No patches are dropped during inference.

Table 1: Open-vocabulary detection performance on LVIS and ODinW. Rows for our models are shown in blue . None of our models have seen any human box annotations for LVIS$_{rare}$ classes at any stage of training, so LVIS AP$^{val}_{rare}$ (rightmost column) measures zero-shot performance. Numbers in green or red indicate the difference to the prior state of the art, i.e. F-VLM R50x64 in the open-vocabulary (top) part of the table and DetCLIPv2 Swin-L in the curated-vocabulary (bottom) part. Gray O+VG indicates that O365+VG were used indirectly (for training the annotator). Gray ODinW numbers indicate that these models were trained on OpenImages data, which overlaps with ODinW. AP$^{mini}$ refers to the LVIS "minival" split introduced by MDETR [14].

| | Method | Backbone | Self-training data | Self-training vocabulary | Human box annotations | ODinW 13 | LVIS AP$^{mini}_{all}$ | LVIS AP$^{mini}_{rare}$ | LVIS AP$^{val}_{all}$ | LVIS AP$^{val}_{rare}$ |
|---|---|---|---|---|---|---|---|---|---|---|
| *Open vocabulary* (evaluation vocabulary is not available at training time): | | | | | | | | | | |
| 1 | RegionCLIP [46] | R50x4 | CC3M | 6k concepts | LVIS$_{base}$ | – | – | – | 32.3 | 22.0 |
| 2 | OWL [26] | CLIP B/16 | – | – | O365+VG | – | – | – | 27.2 | 20.6 |
| 3 | OWL [26] | CLIP L/14 | – | – | O365+VG | 48.4 | – | – | 34.6 | 31.2 |
| 4 | GLIPv2 [45] | Swin-T | Cap4M | tokens | O365+GoldG | 48.5 | 29.0 | – | – | – |
| 5 | GLIPv2 [45] | Swin-B | CC15M | tokens | FiveODs+GoldG | 54.2 | 48.5 | – | – | – |
| 6 | GLIPv2 [45] | Swin-H | CC15M | tokens | FiveODs+GoldG | 55.5 | 50.1 | – | – | – |
| 7 | F-VLM [19] | R50x4 | – | – | LVIS$_{base}$ | – | – | – | 28.5 | 26.3 |
| 8 | F-VLM [19] | R50x64 | – | – | LVIS$_{base}$ | – | – | – | 34.9 | 32.8 |
| 9 | 3Ways [1] | NFNet-F0 | TODO | captions | LVIS$_{base}$ | – | – | – | 35.7 | 25.6 |
| 10 | 3Ways [1] | NFNet-F6 | TODO | captions | LVIS$_{base}$ | – | – | – | 44.6 | 30.1 |
| 11 | OWL-ST | CLIP B/16 | WebLI | N-grams | O+VG | 48.8 | 31.8 | 35.4 | 27.0 | 29.6 -3.2 |
| 12 | OWL-ST | CLIP L/14 | WebLI | N-grams | O+VG | 53.0 | 38.1 | 39.0 | 33.5 | 34.9 +2.1 |
| 13 | OWL-ST | SigLIP G/14 | WebLI | N-grams | O+VG | 49.9 | 37.8 | 40.9 | 33.7 | 37.5 +4.7 |
| 14 | OWL-ST+FT | CLIP B/16 | WebLI | N-grams | O+VG, LVIS$_{base}$ | 48.6 | 47.2 | 37.8 | 41.8 | 36.2 +3.4 |
| 15 | OWL-ST+FT | CLIP L/14 | WebLI | N-grams | O+VG, LVIS$_{base}$ | 50.1 | 54.1 | 46.1 | 49.4 | 44.6 +11.8 |
| 16 | OWL-ST+FT | SigLIP G/14 | WebLI | N-grams | O+VG, LVIS$_{base}$ | 50.1 | 51.3 | 50.9 | 47.0 | 47.2 +14.4 |
| *Human-curated vocabulary* (evaluation vocabulary may be accessed at training time): | | | | | | | | | | |
| 17 | Detic [47] | R50 | IN-21k | LVIS classes | LVIS$_{base}$ | – | – | – | 32.4 | 24.6 |
| 18 | DetCLIPv2 [38] | Swin-T | CC15M | Nouns+curated | O365+GoldG | – | 40.4 | 36.0 | 32.8 | 31.0 |
| 19 | DetCLIPv2 [38] | Swin-L | CC15M | Nouns+curated | O365+GoldG | – | 44.7 | 43.1 | 36.6 | 33.3 |
| 20 | OWL-ST+FT | CLIP B/16 | WebLI | N-grm+curated | O+VG, LVIS$_{base}$ | 48.9 | 51.1 | 41.9 | 45.6 | 40.5 +7.2 |
| 21 | OWL-ST+FT | CLIP L/14 | WebLI | N-grm+curated | O+VG, LVIS$_{base}$ | 48.7 | 55.8 | 50.0 | 50.4 | 45.9 +12.6 |

**Instance selection.** OWL-ViT is an encoder-only architecture and predicts one bounding box per encoder token. This is inefficient, since there are typically many more encoder tokens than objects (e.g. 5184 tokens for resolution $1008 \times 1008$ and patch size $14 \times 14$). Most output tokens therefore do not represent objects. We introduce an *objectness head* which predicts the likelihood that an output token actually represents an object, and compute boxes, class scores, and losses only for the top $k$ tokens by objectness, similar to Efficient DETR [40]. The objectness head receives an encoder token as input and computes a scalar objectness score. The objectness score predicts the future classification score of a token and is supervised by the actual classification score of those tokens that end up being selected and passed on to the classification head. We select approximately 10% of instances by top objectness during training in all of our experiments. During inference, all instances are used.

**Mosaics.** During self-training, we combine raw images into grids of up to $6 \times 6$ to produce a single training example (i.e. a more extreme version of the mosaics in [26]). This has two main motivations: (1) Using mosaics increases the number of raw images seen for a given fixed model input resolution. An alternative is to train using variable image sizes [38], but this would require resizing image position embeddings for each input size. (2) The average resolution and complexity of Web images is lower than images in detection benchmarks and applications. Mosaics reduce the average object size and improve small-object performance, similar to large scale-jittering [8], but with less padding. For all self-training experiments, we use $1 \times 1$, $2 \times 2$, $3 \times 3$, $4 \times 4$, and $6 \times 6$ grids in equal proportions, resulting in an average of 13.2 raw component images per training example.

To further improve training efficiency, we also adopt previously proposed practices for large-scale Transformer training [42] (details in Appendix A.7). Together, our improvements reduce training FLOPS by approximately 50% compared to the original OWL-ViT [26] and increase training throughput by $2\times$ (e.g. for L/14 at $840 \times 840$ resolution measured on TPUv3: GFLOPs/example 11'945.4 vs. 5357.9; examples/s/core 1.0 vs. 2.2). We refer to the improved model as OWLv2.

At inference, no token dropping or instance selection is performed. Inference is therefore identical to the original OWL-ViT, i.e. each image encoder token is decoded into a bounding box and a list of per-query classification scores.

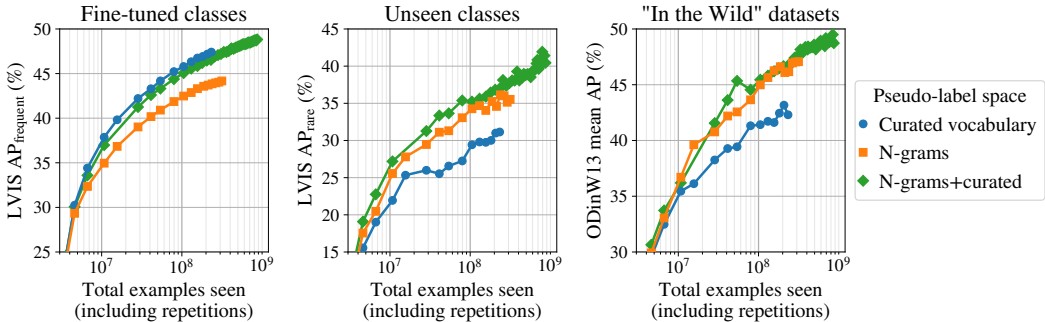

Figure 2: Comparison of pseudo-label spaces. Self-training on a human-curated list of classes yields good downstream performance on these classes, but generalizes poorly to unseen classes and datasets. Open-vocabulary generalization can be improved by obtaining weak but diverse supervision from image-associated text. WebLI image-text data was pseudo-annotated using OWL-ViT CLIP-L/14 with one of three label spaces: *Curated vocabulary* (the union of label spaces from LVIS, Objects365, OpenImagesv4, and Visual Genome), *N-grams* (lightly filtered N-grams from the text associated with each image), or a combination of both (*N-grams + curated*). OWLv2-B/16 models were then self-trained on the pseudo-annotations and fine-tuned on LVIS$_{base}$. Each point represents a separate fine-tuning run. "Examples seen" refers to the number of images after creating mosaics; the total number of raw images seen is $13.2\times$ that number (Section 3.2).

## 3.3 Fine-tuning

Self-training on pseudo-annotations alone already yields strong performance (Section 4.2). However, fine-tuning briefly on human annotations can provide significant further benefits. For fine-tuning, we start with the learning rate and optimizer state of the self-trained checkpoint and then continue training on the target dataset while linearly cooling down the learning rate to zero. Fine-tuning of open-vocabulary models involves a trade-off between improving the performance on the fine-tuned classes and losing open-vocabulary performance [30, 29, 37]. We study this trade-off in Section 4.6.

## 4 Experiments

### 4.1 Experimental Setup

**Models.** We use the publicly available OWL-ViT CLIP L/14 model to generate detection pseudo-annotations for the WebLI dataset (10 billion image-text pairs [4]). For all self-training experiments, we use OWL-ViT models modified as described in Section 3.2. Backbones are initialized with the publicly available CLIP [30] checkpoints (B/16 and L/14) or a SigLIP [43] checkpoint (G/14).

**Training.** Models are first self-trained on the pseudo-annotations for varying durations as indicated. If indicated, after self-training, models are fine-tuned on LVIS$_{base}$, i.e. the LVIS dataset [10] with all annotations for "rare" categories removed. Therefore, neither the annotator nor any of our models have seen human-generated annotations for LVIS$_{rare}$ classes. Fine-tuning uses mosaics up to $3 \times 3$ and is always done until the model has seen 256'000 mosaics (1.1M individual images, roughly equivalent to 100 LVIS epochs). The image size is $960 \times 960$ for /16 models and $1008 \times 1008$ for /14 models. See Appendix A.8 for a complete list of hyperparameters.

**Evaluation.** We use mean average precision (mAP) on LVIS [10] as our main detection metric, where mAP$_{rare}$ indicates open-vocabulary performance on unseen classes. To measure generalization on diverse real-world tasks, we evaluate zero-shot performance on the "Object Detection in the Wild" (ODinW) benchmark [21]. ODinW is a suite of datasets covering a wide range of domains. We report the average mAP on the subset of 13 ODinW datasets introduced in [22] and provide performance on individual datasets in Appendix A.9.2. To avoid leakage of evaluation data into the training set, WebLI was filtered to remove images similar to those in the train, validation, and test splits of 68 common computer vision datasets, including COCO/LVIS, Objects365, and Visual Genome, but not the ODinW datasets (see [4] for details).

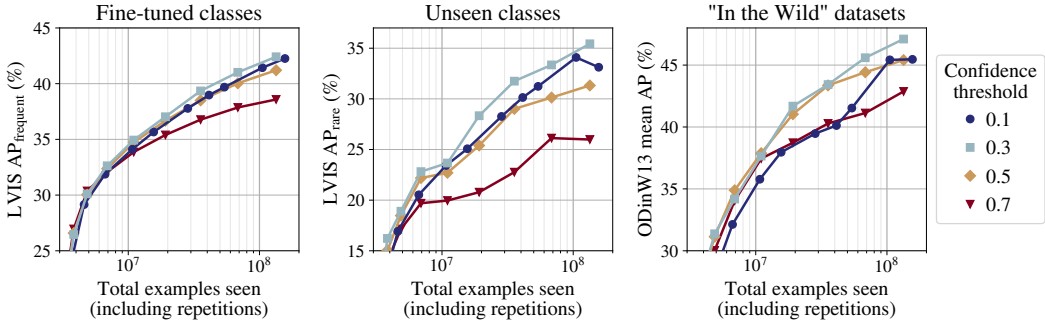

Figure 3: Impact of pseudo-annotation filtering by detection confidence on self-training effectiveness. Pseudo-labels (N-gram label space) were filtered using different confidence thresholds. Number of remaining images for each threshold: 0.1: 5B, 0.3: 2B, 0.5: 782M, 0.7: 224M. OWLv2-B/16 detectors were self-trained on the filtered pseudo-annotations and fine-tuned on $LVIS_{base}$. Each point represents a different fine-tuning run. "Examples seen" refers to the number of images after creating mosaics; the total number of raw images seen is $13.2\times$ that number (Section 3.2).

## 4.2  Main Result

We compare our best models to the literature in Table 1. We broadly include state-of-the-art open-vocabulary detectors in the comparison. Our self-training approach, using only machine-generated pseudo-annotation queries, improves over previous methods even without fine-tuning (Table 1, OWL-ST, rows 11–13). Our OWL-ST B/16 model (row 11) achieves 29.6% LVIS $mAP_{rare}$, 9 points more than the equivalent OWL-ViT model (row 2). Our largest model, G/14 (row 13), reaches 37.5% $mAP_{rare}$, 4.7 points better than the next-best model from the literature (F-VLM R50x64, row 8). Interestingly, after self-training, our models perform *better* on LVIS $mAP_{rare}$ than $mAP_{all}$ (which includes frequent and common classes). We speculate that this may be because weak Web-data supervision may be better for specific terms than general terms: Image/text pairs involving unusual objects (such as $LVIS_{rare}$ categories) may be more likely to be specifically about these objects, whereas common terms like "person" or "car" may occur often without being related to the image.

Fine-tuning on $LVIS_{base}$ provides additional significant improvements, even on $mAP_{rare}$ (OWL-ST+FT, rows 14–16). Our best model, which has only seen machine-generated queries during self-training, reaches 47.2% LVIS $mAP_{rare}$ after fine-tuning, a 14.4-point improvement over the next best model (F-VLM R50x64, row 8).

Including a human-curated list of common object classes as pseudo-annotation queries can further improve the results on LVIS (rows 20–21), but this approach is not fully open-vocabulary since the model sees a curated label space, including the LVIS classes, at training time. While the benefit of the curated label space is significant for our smallest model, is is minor on $mAP_{rare}$ for the larger L/14 model (compare rows 15 and 21).

To measure more general open-world performance, Table 1 also includes zero-shot results on ODinW13 [21], a suite of "in the wild" datasets. Performance on ODinW is best right after self-training and is reduced by fine-tuning on $LVIS_{base}$. We discuss this further in Section 4.6. We also fine-tuned on COCO, where our B/16 and L/14 models reach 54.3% and 56.0% COCO mAP, respectively. OWLv2 therefore matches the performance of ViTDet with a Cascade Mask-RCNN head [23], despite using a simpler head architecture. Further results and examples in Appendix A.9.

## 4.3  Pseudo-Annotation Label Space

Figure 2 takes a closer look at the impact of the pseudo-annotation label space on performance after fine-tuning. Performance on *fine-tuned classes* ($mAP_{frequent}$; left plot) is highest if the pseudo-annotation label space included these classes (blue circles). Therefore, if the target label space is known ahead of time, pseudo-labeling on that space leads to the best results.

However, performance on *unseen classes* ($mAP_{rare}$) and "In the Wild" datasets is much better if the pseudo-labeling included diverse queries that were machine-generated from the image-associated text (orange squares and green diamonds). A mixture of human and machine-generated label spaces

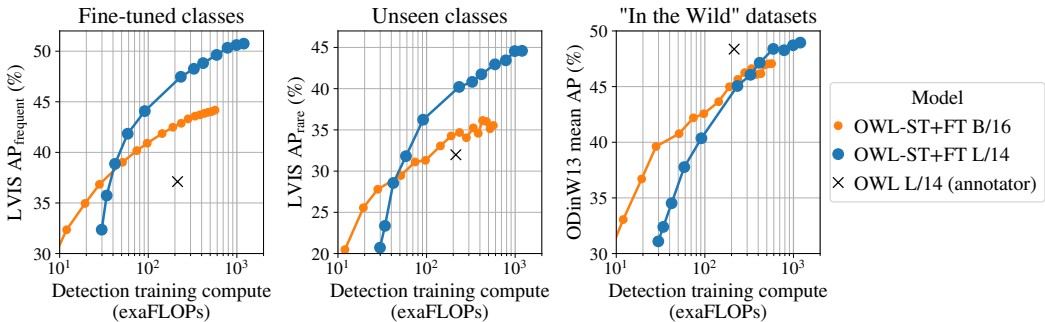

Figure 4: Scaling of detection performance with model size and training compute. Models show classic scaling behavior [42]: Performance increases monotonically with training compute, with larger models being necessary to benefit from larger amounts of compute/data. Models were self-trained on N-gram pseudo-annotations and fine-tuned on $LVIS_{base}$.

performs well in all settings, but does not significantly outperform the purely machine-generated label space on the "In the Wild" datasets. These results suggest that a human-curated label space can help if the target label space is known, but that strong in-the-wild generalization is driven by the weakly supervised machine-generated label space. Our results also show that a simple N-grams approach is sufficient to leverage the weak supervision and outperforms more complex methods (Table 1).

## 4.4 Filtering of Pseudo-Annotations

Besides the label space, a second important decision in self-training is the filtering of pseudo-annotations. We filter based on the detection confidence score of the annotator and vary the score threshold in Figure 3. For confidence-based filtering, a bias-variance trade-off exists between including only high-confidence pseudo-annotations but inheriting the annotator's biases, or lowering the bias but increasing the noise by including lower-confidence pseudo-annotations. Many prior works err on the side of high bias and low variance, applying high confidence thresholds [35] or including only the single highest-confidence detection for an image [47, 1]. In our setting, we find that including all pseudo-annotations that pass a moderate threshold of 0.3 works well, while strict thresholds lead to poor results (Figure 3). As training continues for longer than what was possible for Figure 3, we suspect that lower thresholds may scale better. Therefore, for our main results, we chose to include all annotations above 0.1, but only kept images with at least one annotation above 0.3.

## 4.5 Scaling

The use of abundant Web image-text data with little filtering means that our self-training dataset is large (approximately 2B images). We can therefore study detection training scaling in the same regime as prior work on classification (Figure 4; models see each image at most once for these experiments). We make several noteworthy observations:

1. Self-training is beneficial already at moderate compute budgets, less than that of the annotator.
2. Models show similar scaling behavior for detection as for classification [42]: Both overall performance and the size of the Pareto-optimal model increase with compute/data size.
3. As we move further out of distribution, the amount of compute at which L/14 overtakes B/16 increases. In other words, for in-the-wild performance, at most compute budgets, it may be better to train a smaller model for longer than a larger model for shorter.

These results suggests that self-training on Web data is further scalable as an approach for improving open-vocabulary localization models without the need for further human annotations. The large datasets also makes it possible to scale model size. We trained a G/14 model, which has $5.2\times$ the number of parameters and $4.3\times$ the inference FLOPs of our L/14 model. To our knowledge, this is the largest open-vocabulary detection model to date. Since the G/14 model uses a different backbone than our other models (SigLIP [43] instead of CLIP [30]), we do not include it in Figure 4, but show in Table 1 that it is currently the best-performing model on zero-shot LVIS, with 47.2% $mAP_{rare}$.

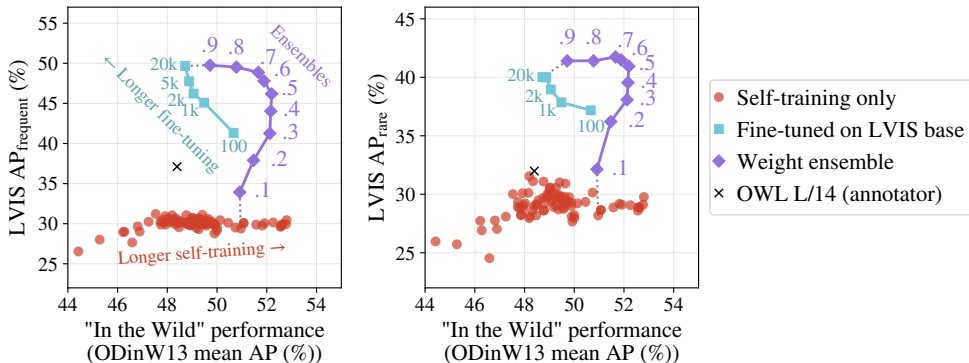

Figure 5: Trade-off between fine-tuned and open-world performance. Self-training yields continued improvements on a suite of diverse datasets (ODinW13; $x$-axis), but performance on any given dataset (e.g. LVIS; $y$-axis) may saturate (red circles). Fine-tuning on a target dataset improves performance on that dataset, but reduces the open-world generalization ability in proportion to the finetuning duration (light blue squares; numbers indicate finetuning steps). This trade-off can be improved through weight-space ensembling (averaging) of the pretrained and fine-tuned checkpoints [37] (purple diamonds; numbers indicate the mixing coefficient for the fine-tuned weights). The plot shows B/16 models self-trained on N-gram pseudo-annotations and evaluated either directly after self-training or after fine-tuning on LVIS$_{base}$. Ensembles were created between the longest-self-trained checkpoint and the weights obtained after finetuning that checkpoint for 20k steps. Note that there is significant variability in ODinW13 performance between checkpoints towards the end of self-training.

## 4.6 Effect of Fine-Tuning on Open-Vocabulary Performance

For contrastively trained image-text models, fine-tuning improves performance on the target distribution but reduces the (originally very high) robustness to distribution shift [30, 28, 37]. We observe the same effect for detection, using ODinW13 AP as a proxy for out-of-distribution performance: Compared to the performance after self-training (red dots in Figure 5), fine-tuning on LVIS$_{base}$ improves performance on the fine-tuned classes (LVIS mAP$_{frequent}$), but OOD performance (ODinW13 AP) is simultaneously reduced in proportion to the amount of fine-tuning (light blue line in Figure 5).

A simple approach to improve on this trade-off is to create an ensemble of the model before and after fine-tuning by averaging the model weights [37]. This approach comes at no additional training cost and improves the Pareto-frontier for all ensemble mixing ratios (Figure 5, purple line). We also tried co-training on WebLI and LVIS but found it to perform worse than weight ensembling.

Notably, performance on LVIS$_{rare}$ behaves similarly to LVIS$_{frequent}$ and *improves* during fine-tuning, even though no LVIS$_{rare}$ classes are seen (Figure 5, right). This may be because LVIS$_{rare}$ classes are semantically and visually close to LVIS$_{frequent}$ classes. For example, seeing many annotations for `"bird"` may improve performance on rare classes such as `"heron"`, `"mallard"`, or `"puffin"`. LVIS mAP$_{rare}$ therefore only measures a narrow concept of open-vocabulary performance, and does not reveal the fact that fine-tuning significantly *reduces* generalization to broader distribution shifts. Benchmarks such as ODinW therefore provide significant additional insight.

# 5 Limitations

The main limitation of our method is the amount of compute and data needed for self-training. As we show in Section 4.5, performance improves consistently with training compute and data. This means that further improvements are possible, but also that these will come at increasingly large costs. In fact, cost likely increases faster than resources can realistically be grown in practice. New approaches will therefore be eventually necessary for further improvements.

A second important limitation of our method, similar to other open-vocabulary models [30, 28, 37], is the trade-off between fine-tuned and open-vocabulary performance addressed in Section 4.6. For out-of-distribution queries, predictions of fine-tuned models may be poorly calibrated and may depend on the precise wording of the query. These issues can be mitigated with weight ensembling [37], but more research is needed to fully understand the open-vocabulary robustness of these models.

# 6 Conclusion

In the past, open-vocabulary detection performance has been limited by the availability of human-annotated detection training data. Here, we show that self-training can be scaled up to overcome the dependency on human annotations. Our OWL-ST recipe delivers large improvements in detection performance using weak supervision from abundant Web data, similar to what has been seen for image classification and language modelling.

## Acknowledgments and Disclosure of Funding

We would like to thank Xiao Wang for help with the WebLI dataset, Xiaohua Zhai and Lucas Beyer for providing the SigLIP model, and Rich Munoz and Alexey Dosovitskiy for insightful comments.

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

# A Appendix

The Appendix provides a Model Card [27] for OWLv2 as well as additional methodological details, hyperparameters, and results. At the end of the Appendix, we provide qualitative examples of the self-training data and model predictions.

The Appendix is structured as follows:

## A.1 Model Card

| Model Summary | |
|---|---|
| Model Architecture | OWL v2 is an open-vocabulary object detector based on OWL-ViT [26]. It consists of an image encoder with a Vision Transformer [17] architecture, a text encoder with a similar Transformer architecture, and heads that predict bounding boxes and label scores from provided images and text queries. |
| Input(s) | An image and a list of free-text object descriptions (queries). |
| Output(s) | A list of bounding boxes and a score for each box/query pair. |
| **Usage** | |
| Application | The model is intended for open-vocabulary object detection. |
| Known Caveats | (1) Confidence scores of predictions are not intended to be compared across text queries. While the training loss encourages cross-query calibration for *seen* queries, scores for *unseen* queries are not calibrated. Further, the mean Average Precision (mAP) metric does not measure cross-query calibration, so higher mAP does not imply better cross-query calibration. Also see Section 5. (2) Fine-tuning the model creates a trade-off between the performance on fine-tuned texts and unseen texts. See Section 4.6 for details. |
| **System Type** | |
| System Description | This is a standalone model. |
| Upstr. Dependencies | None. |
| Downstr. Dependencies | None. |
| **Implementation Frameworks** | |
| Hardware & Software | Hardware: TPU [13] v2 or v3 (for B- and L-sized models) or v4 (for G-sized models). Software: JAX [3], Flax [11], Scenic [7]. |
| Compute Requirements | Reported in Section 4.5. |
| **Model Characteristics** | |
| Model Initialization | The model is initialized from pre-trained language CLIP [30] or SigLIP [43] checkpoints. |
| Model Status | This is a static model trained on an offline dataset. |
| Model Stats | The largest OWLv2 model has 2.3B parameters, of which 2B are used for the image encoder and 300M for the text encoder (the heads have a negligible number of parameters). We also trained models with 430M and 150M parameters. |
| **Data Overview** | |
| Training dataset | The model is self-trained on bounding boxes predicted by the original OWL-ViT L/14 model [26] on the WebLI dataset [4]. Details on the annotation procedure are provided in Section 3.1. |
| Evaluation & Fine-tuning Dataset | Open-vocabulary object detection performance is evaluated using the LVIS [10] and ODinW13 [21] datasets. As indicated in Table 1, some models are fine-tuned on the "base" annotations of LVIS, i.e. only annotations for "frequent" and "common" object categories as defined in the official annotations [10]. None of our models have seen any human annotations for LVIS "rare" categories, such that LVIS $mAP_{rare}$ measures zero-shot performance. |
| **Evaluation Results** | |
| Evaluation Results | Reported in Table 1. |
| **Model Usage & Limitations** | |
| Sensitive Use | The model detects objects matching free-text descriptions. This capability should not be used for unethical use cases such as surveillance. |
| Known Limitations | Reported in Section 5. |
| Ethical Considerations | Reported in Section 5. |

## A.2 Human-Curated Label Space

The human-curated label space was obtained by merging common dataset class lists with the Python code below.

```python
# Dataset class names, as available e.g. from TensorFlow Datasets.
# For Visual Genome, we used the 1600 most common label strings.
LVIS_CLASS_NAMES = [...]
OBJECTS365_CLASS_NAMES = [...]
OPEN_IMAGES_V4_BOXABLE_CLASS_NAMES = [...]
VISUAL_GENOME_CLASS_NAMES = [...]

queries = (
    LVIS_CLASS_NAMES
    + OBJECTS365_CLASS_NAMES
    + OPEN_IMAGES_V4_BOXABLE_CLASS_NAMES
)

# Remove duplicates:
queries = set([q.lower() for q in queries])

# Remove plural forms:
remove = set()
for singular in queries:
  plurals = [singular + 's', singular + 'es']
  for plural in plurals:
    if plural in queries:
      remove.add(plural)

# Same queries for all images:
queries = list(queries.difference(remove))
```

## A.3 Machine-Generated Label Space

The machine-generated label space was obtained from the image-associated text, for each image separately, using the Python code below. Figure A3 shows example pseudo-annotations using the N-gram label space.

```python
from typing import Iterable, List
import nltk

# Stopwords from nltk.corpus.stopwords.words('english'):
STOPWORDS_EN = frozenset({
    'a', 'about', 'above', 'after', 'again', 'against', 'all', 'am', 'an',
    'and', 'any', 'are', 'as', 'at', 'be', 'because', 'been', 'before', 'being',
    'below', 'between', 'both', 'but', 'by', 'can', 'did', 'do', 'does',
    'doing', 'don', 'down', 'during', 'each', 'few', 'for', 'from', 'further',
    'had', 'has', 'have', 'having', 'he', 'her', 'here', 'hers', 'herself',
    'him', 'himself', 'his', 'how', 'i', 'if', 'in', 'into', 'is', 'it', 'its',
    'itself', 'just', 'me', 'more', 'most', 'my', 'myself', 'no', 'nor', 'not',
    'now', 'of', 'off', 'on', 'once', 'only', 'or', 'other', 'our', 'ours',
    'ourselves', 'out', 'over', 'own', 's', 'same', 'she', 'should', 'so',
    'some', 'such', 't', 'than', 'that', 'the', 'their', 'theirs', 'them',
    'themselves', 'then', 'there', 'these', 'they', 'this', 'those', 'through',
    'to', 'too', 'under', 'until', 'up', 'very', 'was', 'we', 'were', 'what',
    'when', 'where', 'which', 'while', 'who', 'whom', 'why', 'will', 'with',
    'you', 'your', 'yours', 'yourself', 'yourselves'
})

# These words were found by manually going through the most common 1000 words
# in a sample of alt-texts and selecting generic words without specific meaning:
COMMON_GENERIC_WORDS = frozenset({
    'alibaba', 'aliexpress', 'amazon', 'available', 'background', 'blog', 'buy',
```

```
26      'co', 'com', 'description', 'diy', 'download', 'facebook', 'free', 'gif',
27      'hd', 'ideas', 'illustration', 'illustrations', 'image', 'images', 'img',
28      'instagram', 'jpg', 'online', 'org', 'original', 'page', 'pdf', 'photo',
29      'photography', 'photos', 'picclick', 'picture', 'pictures', 'png', 'porn',
30      'premium', 'resolution', 'royalty', 'sale', 'sex', 'shutterstock', 'stock',
31      'svg', 'thumbnail', 'tumblr', 'tumgir', 'twitter', 'uk', 'uploaded', 'vector',
32      'vectors', 'video', 'videos', 'wallpaper', 'wallpapers', 'wholesale', 'www',
33      'xxx', 'youtube'
34  })
35
36  def _is_all_stopwords(ngram: Iterable[str]) -> bool:
37      return set(ngram).issubset(STOPWORDS_EN)
38
39
40  def _get_ngrams(
41      caption: str, max_num_queries: int, max_ngram_len: int
42  ) -> List[str]:
43      """Returns image caption ngrams as queries."""
44
45      # Make lower-case:
46      caption = caption.lower()
47
48      # Remove common generic words:
49      words = [w for w in caption.split() if w not in COMMON_GENERIC_WORDS]
50
51      queries = []
52      for ngram in nltk.everygrams(words, max_len=max_ngram_len):
53          # Don't use ngram if it only consists of stop words:
54          if _is_all_stopwords(ngram):
55              continue
56          queries.append(' '.join(ngram))
57          if len(queries) == max_num_queries:
58              break
59      return queries
60
61  # Example command to get queries for one image:
62  queries = _get_ngrams(caption, max_num_queries=300, max_ngram_len=10)
```

## A.4   Combined Label Space

When merging pseudo-annotations obtained with human-curated and machine-generated queries, it is important to consider that human-curated queries tend to be closer to the training distribution of the annotator and therefore tend to have higher scores than pseudo-annotations based on machine-generated queries. Simply merging annotations from the two label spaces and filtering them with the same confidence threshold would therefore retain primarily annotations based on human-curated queries. To achieve a more even balance when using the combined label space ("N-grm+curated" in Table 1), we therefore re-scaled scores of pseudo-annotations obtained with the human-curated queries by a factor of 0.3 before applying the same confidence threshold to all (human-curated and machine-generated) annotations.

## A.5   Augmentations for Self-Training

Since Web-scale image-text data differs in important aspects from human-curated detection datasets, we depart from the augmentation strategy of [26] in several ways. As described in Section 3.2, since Web images tend to be smaller and show fewer objects than e.g. LVIS images, we use stronger image mosaics with up do $6 \times 6$ tiles (Figure A1). For the same reason, we additionally randomly resize each raw image such that its width is between $0.5\times$ and $1.0\times$ the width of the full mosaic tile, padding on the bottom and right to preserve the aspect ratio (Figure A4).

On the other hand, given the large size of our dataset, some other augmentations can be avoided: We do not use left/right flipping or random cropping during self-training. We also do not add random

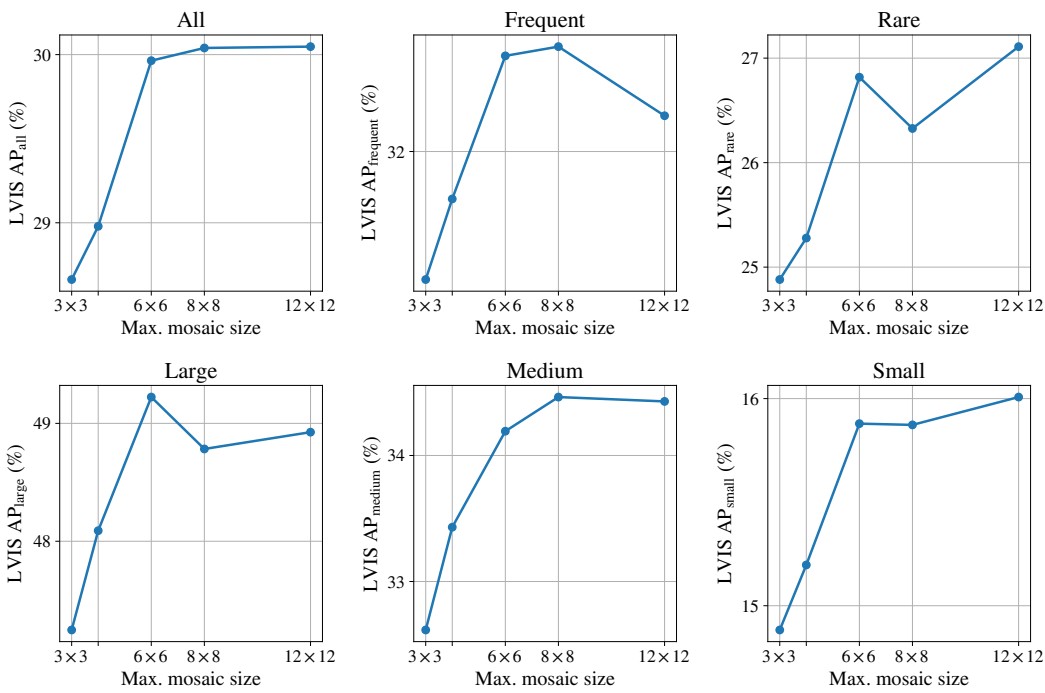

Figure A1: Sweep over mosaic sizes. OWL-ViT B/16 models were trained on pseudo-box annotations ("ngrams" label space) for 100'000 steps with different mosaic sizes. At a given "max. mosaic size", the model is trained on equal proportions of mosaics up to that size. For example, for max. size $= 12 \times 12$, the model receives images with $1$, $2^2$, $3^2$, $4^2$, $6^2$, $8^2$, or $12^2$ tiles, respectively (only sizes with prime factors 1, 2, and 3 are supported). For this figure, the model input resolution was $768 \times 768$. Mosaic sizes up to $12 \times 12$ improve overall performance ($\text{mAP}_{\text{all}}$) and especially "rare" and "small" object performance. The benefit may be due to seeing smaller objects on average, or due to seeing more WebLI images per training step (a $12 \times 12$ mosaic contains 144 WebLI images).

prompt templates to the pseudo-labels during self-training. During fine-tuning, we use the same augmentations as [26].

### A.6  Token Dropping

To improve training efficiency, we drop image patches based on their pixel variance (Section 3.2). Table A2 shows how the performance of a standard OWL-ViT model varies for different amounts of token dropping. Dropping up to 50% of tokens is within one standard deviation of the full performance. We therefore drop 50% of tokens during all of our experiments.

Table A2: Performance of standard OWL-ViT (L/14), trained on Objects365 and Visual Genome as in [26], for different token drop rates. For drop rate 0.0, the standard deviation over three runs is given.

| Metric | Token drop rate | | | | |
| --- | --- | --- | --- | --- | --- |
| | 0.00 | 0.25 | 0.33 | 0.50 | 0.70 |
| LVIS $\text{AP}^{\text{val}}_{\text{all}}$ | 33.3 ±0.33 | 33.1 | 33.6 | 32.9 | 30.4 |
| LVIS $\text{AP}^{\text{val}}_{\text{rare}}$ | 31.8 ±1.16 | 31.0 | 32.6 | 30.8 | 28.2 |

To inject some stochasticity to the patch selection, we add a small amount of noise to the image before computing patch variance (uniformly distributed between 0.0 and 0.01 for images in the range $[0.0, 1.0]$). Figure A4 shows an example training image before and after token dropping.

Table A3: Hyperparameters of the models shown in Table 1. Only parameters that vary between models are shown; constant parameters are described in the text (Appendix A.8). For *Dropout rate* and *Droplayer rate*, the first number indicates the value used for the image encoder and the second for the text encoder. *Examples seen* includes both self-training and fine-tuning.

| Method | Backbone | Image size | Learning rate | Dropout rate | Droplayer rate | Instance top $k$ | Batch size (ST) | Batch size (FT) | Examples seen |
|---|---|---|---|---|---|---|---|---|---|
| *Open vocabulary:* | | | | | | | | | |
| 11 OWL-ST | CLIP B/16 | 960 | $5 \times 10^{-5}$ | .0/.0 | .2/.1 | 256 | 256 | – | $3.7 \times 10^8$ |
| 12 OWL-ST | CLIP L/14 | 1008 | $2 \times 10^{-5}$ | .0/.0 | .2/.1 | 512 | 256 | – | $2.3 \times 10^8$ |
| 13 OWL-ST | SigLIP G/14 | 1008 | $2 \times 10^{-5}$ | .0/.1 | .2/.4 | 512 | 128 | – | $1.6 \times 10^8$ |
| 14 OWL-ST+FT | CLIP B/16 | 960 | $5 \times 10^{-5}$ | .0/.0 | .2/.1 | 256 | 256 | 256 | $3.6 \times 10^8$ |
| 15 OWL-ST+FT | CLIP L/14 | 1008 | $2 \times 10^{-5}$ | .0/.0 | .2/.1 | 512 | 256 | 128 | $2.3 \times 10^8$ |
| 16 OWL-ST+FT | SigLIP G/14 | 1008 | $2 \times 10^{-5}$ | .0/.1 | .2/.4 | 512 | 128 | 128 | $1.6 \times 10^8$ |
| *Human-curated vocabulary:* | | | | | | | | | |
| 20 OWL-ST+FT | CLIP B/16 | 960 | $5 \times 10^{-5}$ | .0/.0 | .2/.1 | 256 | 256 | 256 | $8.2 \times 10^8$ |
| 21 OWL-ST+FT | CLIP L/14 | 1008 | $2 \times 10^{-5}$ | .0/.0 | .2/.1 | 512 | 256 | 128 | $3.6 \times 10^8$ |

## A.7  Further Efficiency Improvements

To further improve training efficiency beyond the methods described in Section 3.2, we also adopt previously proposed methods for large-scale Transformer training: To save memory, we use a variant [42] of the Adafactor optimizer [34] instead of Adam [15]. To avoid having to choose and optimize the total training duration ahead of time, we use the open-ended inverse square-root schedule [36, 42] with a fixed time-scale of 10'000 steps for all experiments and linearly "cool down" checkpoints along the way for evaluation (see Section 3.3).

## A.8  Model Hyperparameters

We use the following hyperparameters for all of our models. Hyperparameters that vary between models are listed in Table A3.

- Optimizer: Adafactor variant as in [42]
- Learning rate schedule: Inverse square-root [36] with timescale 10'000 steps
- Learning rate for the text encoder: $2 \times 10^{-6}$
- Token dropping rate during training: 0.5
- Pseudo-annotation confidence score threshold: 0.3 (except for Figure 3)
- Augmentations: See Appendix A.5
- All remaining hyperparameters are as in [26].

**Hyperparameter selection.** Most hyperparameters were either taken directly from [26] or technically constrained, e.g. we chose the largest batch size that fit into the memory of the available accelerators. Where hyperparameters were tuned, we ran short B/16-scale trial experiments and selected the parameters with the highest LVIS mAP$_{\text{rare}}$ for our main runs.

**SigLIP G/14.** For the G/14 model, we started self-training with a learning rate of $5 \times 10^{-5}$, a droplayer rate of .1/.0, and no dropout. We found that the model overfit during fine-tuning with these settings, and switched to a learning rate of $2 \times 10^{-5}$, a droplayer rate of .2/.4, and a dropout rate of .0/.1 after 740'000 self-training steps. To save resources, we did not start training from the beginning. With the new settings, we observed no overfitting during fine-tuning, but it is possible that these settings are still not optimal.

Table A4: Open-vocabulary detection results on LVIS using the "fixed" AP metric [5]. Fixed AP is implemented as proposed in [5] by evaluating AP on the top 10'000 predictions per class over the entire validation set.

| Method | Backbone | $AP_{all}^{mini}$ | | $AP_{rare}^{mini}$ | | $AP_{all}^{val}$ | | $\mathbf{AP_{rare}^{val}}$ | |
|---|---|---|---|---|---|---|---|---|---|
| | | old | fixed | old | fixed | old | fixed | old | fixed |
| *Open vocabulary:* | | | | | | | | | |
| 1  RegionCLIP [46] | R50x4 | – | – | – | – | 32.3 | – | 22.0 | – |
| 2  OWL [26] | CLIP B/16 | – | – | – | – | 27.2 | – | 20.6 | – |
| 3  OWL [26] | CLIP L/14 | – | – | – | – | 34.6 | – | 31.2 | – |
| 4  GLIPv2 [45] | Swin-T | 29.0 | – | – | – | – | – | – | – |
| 5  GLIPv2 [45] | Swin-B | 48.5 | – | – | – | – | – | – | – |
| 6  GLIPv2 [45] | Swin-H | 50.1 | – | – | – | – | – | – | – |
| 7  F-VLM [19] | R50x4 | – | – | – | – | 28.5 | – | 26.3 | – |
| 8  F-VLM [19] | R50x64 | – | – | – | – | 34.9 | – | 32.8 | – |
| 9  3Ways [1] | NFNet-F0 | – | – | – | – | 35.7 | – | 25.6 | – |
| 10  3Ways [1] | NFNet-F6 | – | – | – | – | 44.6 | – | 30.1 | – |
| 11  OWL-ST | CLIP B/16 | 31.8 | 34.4 | 35.4 | 38.3 | 27.0 | 28.6 | 29.6 | 30.3 |
| 12  OWL-ST | CLIP L/14 | 38.1 | 40.9 | 39.0 | 41.5 | 33.5 | 35.2 | 34.9 | 36.2 |
| 13  OWL-ST | SigLIP G/14 | 37.8 | – | 40.9 | – | 33.7 | – | 37.5 | – |
| 14  OWL-ST+FT | CLIP B/16 | 47.2 | 48.7 | 37.8 | 42.1 | 41.8 | 43.2 | 36.2 | 39.0 |
| 15  OWL-ST+FT | CLIP L/14 | 54.1 | 56.2 | 46.1 | 52.3 | 49.4 | 51.1 | 44.6 | 47.4 |
| 16  OWL-ST+FT | SigLIP G/14 | 51.3 | – | 50.9 | – | 47.0 | – | 47.2 | – |
| *Human-curated vocabulary:* | | | | | | | | | |
| 17  Detic [47] | R50 | – | – | – | – | 32.4 | – | 24.6 | – |
| 18  DetCLIPv2 [38] | Swin-T | – | 40.4 | – | 36.0 | – | 32.8 | – | 31.0 |
| 19  DetCLIPv2 [38] | Swin-L | – | 44.7 | – | 43.1 | – | 36.6 | – | 33.3 |
| 20  OWL-ST+FT | CLIP B/16 | 51.1 | 52.3 | 41.9 | 46.5 | 45.6 | 46.7 | 40.5 | 42.5 |
| 21  OWL-ST+FT | CLIP L/14 | 55.8 | 57.2 | 50.0 | 54.5 | 50.4 | 52.0 | 45.9 | 48.5 |

## A.9 Additional Results

### A.9.1 Fixed Average Precision

In the standard Average Precision metric ($AP^{old}$), performance on one class depends on the performance on other classes. This dependence makes the metric "gameable" by re-scaling the scores of certain classes [5]. To avoid this issue, some prior work reports a "fixed" version of AP proposed in [5]. In Table 1, we report $AP^{old}$ for our models. For models from the literature, we report whichever AP version is available. Since $AP^{fixed}$ tends to produce higher values than $AP^{old}$, Table 1 tends to underestimate the advantage of our method over prior work using $AP^{fixed}$. We provide $AP^{fixed}$ for all of our models in Table A4. As proposed in [5], we implement $AP^{fixed}$ by evaluating AP on the top 10'000 predictions per class over the entire validation set. This ensures that classes do not compete with each other for inclusion in the evaluated predictions.

### A.9.2 Per-Dataset ODinW Results

Table A5 shows un-aggregated results on all 35 ODinW datasets for our main models. In addition, in the last row, we provide results for a weight-space ensemble of a self-trained and fine-tuned OWLv2 L/14 model (the same model is shown in Figure A2).

### A.9.3 Fine-Tuning Robustness Trade-Off for OWLv2 L/14

In Figure A2, we provide the same analysis of the robustness trade-off after fine-tuning for an L/14 model that we provided for a B/16 model in Figure 5.

### A.10 Qualitative Examples

In Figures A5 to A7, we provide qualitative examples of detection predictions from OWLv2 L/14 models. In each figure, the top image shows predictions obtained directly after self-training, and

the bottom image shows predictions after fine-tuning on $\text{LVIS}_{\text{base}}$. Example images are from the LVIS validation set and the model was queried with all LVIS classes. All predictions meeting the confidence threshold specified in the caption are shown.

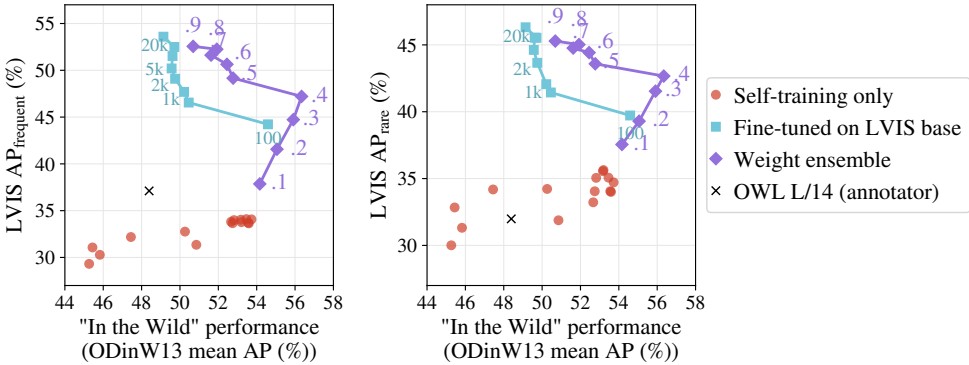

Figure A2: Trade-off between fine-tuned and open-world performance. Similar to Figure 5, but for OWLv2 L/14.

Table A5: Zero-shot AP of the models in Table 1 on all 35 ODinW datasets [21]. The subset of 13 datasets defined in [22] and used in the main paper is shown in **bold**. The last row (*OWL-ST/FT ens*) shows the weight-space ensemble [37] of the checkpoints after self-training and after fine-tuning of the model in row 21 (weight of the fine-tuned checkpoint in the ensemble is 0.4; also see Figure A2). This is our best model by ODinW13 performance.

| | Method | Backbone | Mean (13 datasets) | Mean (35 datasets) | Median (35 datasets) | **Aerial Maritime Drone Large** | Aerial Maritime Drone Tiled | American Sign Language | **Aquarium** | BCCD | Boggle Boards | Brackish Underwater | Chess Pieces | **Cottontail Rabbits** | Dice Medium Color | Drone Control | **Ego Hands Generic** | Ego Hands Specific | Hard Hat Workers | Mask Wearing | Mountain Dew Commercial | **North America Mushrooms** | Open Poetry Vision | Oxford Pets By Breed | Oxford Pets By Species | **Packages** | **Pascal VOC** | **Pistols** | Pk Lot | Plantdoc | **Pothole** | **Raccoon** | Selfdriving Car | **Shellfish OpenImages** | Thermal Cheetah | **Thermal Dogs And People** | Uno Cards Raw | **Vehicles OpenImages** | Website Screenshots | Wildfire Smoke |
|---|---|---|---|---|---|---|---|---|---|---|---|---|---|---|---|---|---|---|---|---|---|---|---|---|---|---|---|---|---|---|---|---|---|---|---|---|---|---|---|---|
| *Open vocabulary:* | | | | | | | | | | | | | | | | | | | | | | | | | | | | | | | | | | | | | | | | |
| 11 | OWL-ST | CLIP B/16 | 48.8 | 22.1 | 11.6 | 11.6 | 19.4 | 1.1 | 33.2 | 11.6 | 0.3 | 4.8 | 4.1 | 85.5 | 0.1 | 2.7 | 46.9 | 5.5 | 2.0 | 0.4 | 22.0 | 33.9 | 0.4 | 2.7 | 3.4 | 75.9 | 52.7 | 60.1 | 0.1 | 4.8 | 19.2 | 66.6 | 5.5 | 40.1 | 19.1 | 51.1 | 1.0 | 57.2 | 1.8 | 25.4 |
| 12 | OWL-ST | CLIP L/14 | 53.0 | 24.4 | 16.2 | 19.9 | 21.2 | 1.1 | 32.3 | 16.2 | 0.2 | 5.9 | 7.8 | 84.9 | 0.1 | 4.7 | 47.1 | 3.5 | 1.9 | 0.5 | 27.3 | 76.6 | 0.6 | 3.1 | 2.7 | 70.9 | 53.9 | 62.6 | 0.0 | 4.4 | 27.5 | 63.8 | 4.9 | 35.0 | 25.5 | 55.6 | 1.1 | 58.5 | 1.8 | 31.1 |
| 13 | OWL-ST | SigLIP G/14 | 49.9 | 22.9 | 17.5 | 22.0 | 17.5 | 2.0 | 36.7 | 21.4 | 0.2 | 3.3 | 5.6 | 88.1 | 0.1 | 4.9 | 37.8 | 4.3 | 1.4 | 0.2 | 22.6 | 42.4 | 0.5 | 3.0 | 3.2 | 62.8 | 53.4 | 58.4 | 0.1 | 6.5 | 25.7 | 63.9 | 5.8 | 42.5 | 25.0 | 56.6 | 1.2 | 58.1 | 2.0 | 23.4 |
| 14 | OWL-ST+FT | CLIP B/16 | 48.6 | 20.8 | 6.0 | 13.7 | 16.6 | 0.2 | 35.8 | 3.9 | 0.1 | 4.2 | 3.1 | 85.5 | 0.1 | 0.9 | 50.7 | 1.3 | 2.7 | 0.5 | 16.0 | 37.4 | 0.2 | 1.9 | 2.1 | 71.3 | 57.4 | 59.4 | 0.2 | 2.7 | 7.6 | 61.7 | 6.0 | 42.5 | 15.3 | 45.6 | 1.3 | 62.8 | 1.2 | 15.8 |
| 15 | OWL-ST+FT | CLIP L/14 | 50.1 | 22.3 | 6.3 | 20.6 | 16.3 | 0.2 | 37.4 | 4.0 | 0.1 | 5.1 | 5.6 | 83.4 | 0.1 | 4.8 | 58.5 | 2.2 | 2.1 | 0.6 | 28.5 | 42.2 | 0.3 | 2.5 | 1.9 | 65.5 | 58.9 | 63.7 | 0.2 | 1.5 | 9.1 | 57.2 | 6.3 | 43.0 | 24.7 | 47.7 | 1.3 | 64.3 | 1.8 | 20.3 |
| 16 | OWL-ST+FT | SigLIP G/14 | 50.1 | 22.5 | 9.5 | 21.3 | 16.5 | 0.3 | 39.8 | 9.5 | 0.3 | 5.6 | 5.8 | 82.5 | 0.0 | 3.6 | 50.9 | 0.5 | 1.7 | 0.2 | 25.5 | 44.9 | 0.2 | 2.8 | 2.3 | 68.1 | 56.4 | 58.5 | 0.7 | 5.3 | 17.4 | 58.3 | 6.1 | 42.7 | 23.6 | 47.9 | 1.9 | 61.9 | 1.9 | 23.9 |
| *Human-curated vocabulary:* | | | | | | | | | | | | | | | | | | | | | | | | | | | | | | | | | | | | | | | | |
| 20 | OWL-ST+FT | CLIP B/16 | 48.9 | 21.7 | 6.8 | 16.7 | 17.2 | 0.3 | 35.3 | 4.5 | 0.1 | 4.6 | 4.4 | 85.1 | 0.1 | 2.4 | 51.8 | 0.9 | 2.9 | 0.4 | 27.3 | 36.9 | 0.3 | 2.1 | 2.5 | 71.3 | 59.0 | 61.3 | 0.4 | 2.7 | 9.6 | 58.7 | 6.8 | 42.0 | 20.0 | 45.7 | 1.2 | 62.6 | 1.5 | 20.6 |
| 21 | OWL-ST+FT | CLIP L/14 | 48.7 | 21.9 | 7.0 | 18.8 | 17.5 | 0.2 | 36.4 | 5.3 | 0.1 | 5.4 | 5.7 | 85.1 | 0.1 | 4.9 | 53.9 | 2.5 | 2.2 | 0.3 | 28.8 | 41.2 | 0.3 | 2.4 | 2.1 | 61.1 | 59.2 | 65.7 | 0.1 | 1.8 | 9.5 | 57.9 | 7.0 | 44.0 | 23.8 | 36.8 | 0.9 | 63.2 | 1.6 | 20.7 |
| | OWL-ST/FT ens | CLIP L/14 | 56.3 | 25.6 | 10.6 | 21.7 | 20.0 | 1.0 | 39.1 | 10.6 | 0.2 | 7.6 | 7.0 | 87.0 | 0.0 | 6.1 | 53.1 | 3.2 | 2.1 | 0.3 | 31.3 | 80.6 | 0.4 | 3.1 | 2.9 | 66.3 | 61.8 | 66.2 | 0.1 | 4.0 | 26.0 | 65.4 | 6.2 | 45.1 | 24.1 | 56.7 | 1.1 | 63.3 | 1.9 | 30.9 |

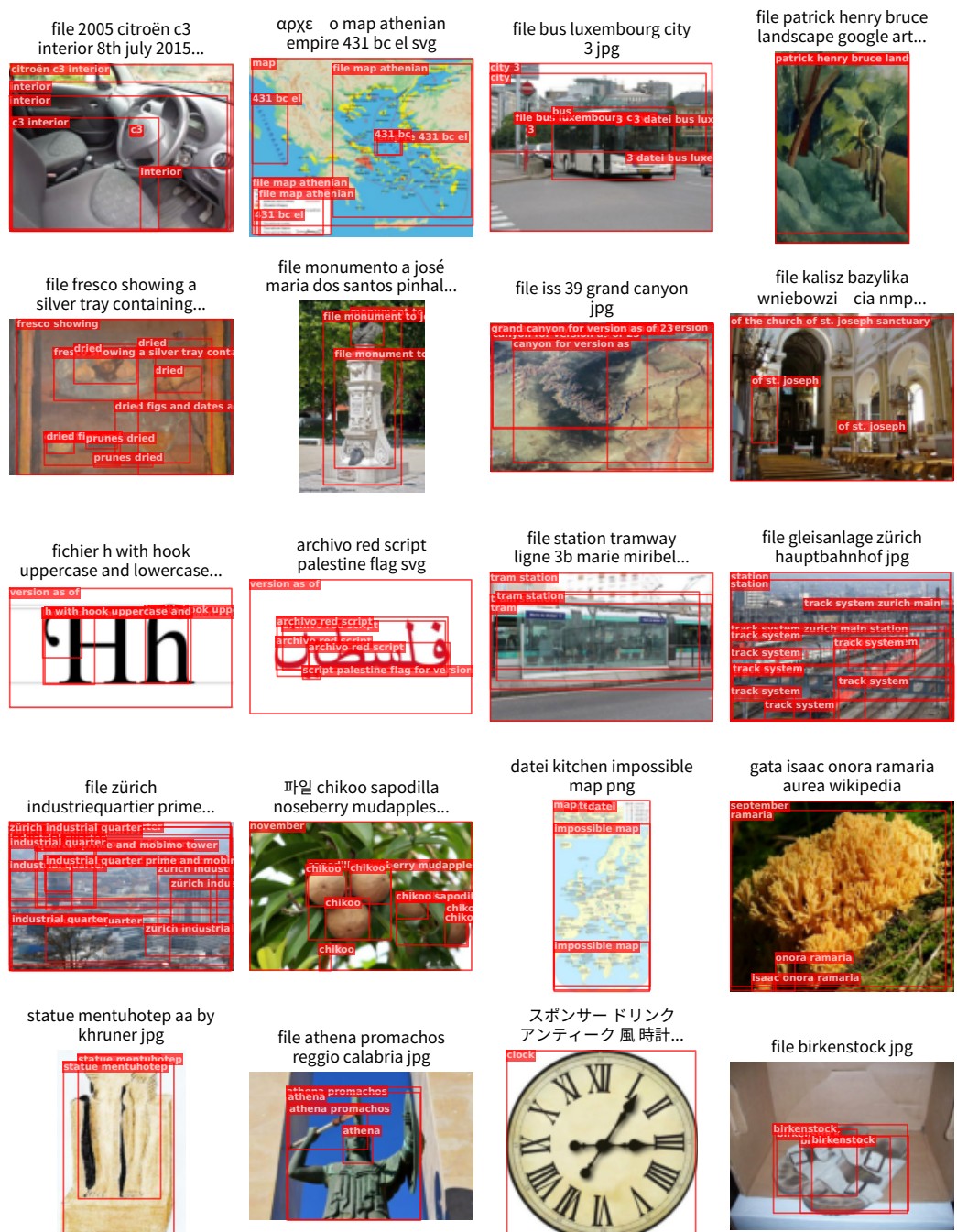

Figure A3: Example pseudo-annotations on WebLI [4]. Image-associated text (from the HTML `alt_text` tag) is shown above the images. If the text is not in English, an automatically generated translation is used. N-grams are extracted from these texts to generate queries for the annotator model. Pseudo-annotations were filtered as for our main experiments: To be included, boxes must have a score of at least 0.1, and images must have at least one box with a score above 0.3. All images from Wikimedia Commons.

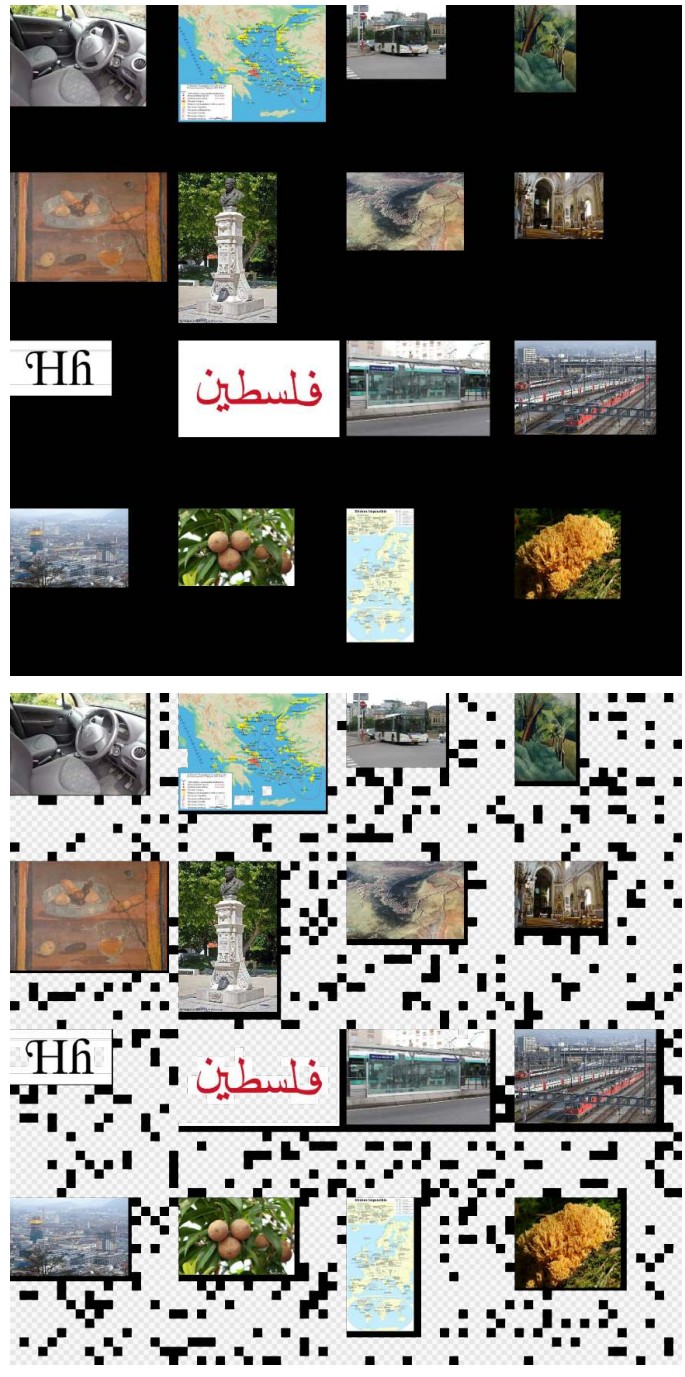

Figure A4: Training inputs after pre-processing. **Top:** A $4 \times 4$ mosaic of randomly resized and padded images as used for self-training. **Bottom:** The same mosaic after dropping the 50% of patches with lowest pixel variance (image size: $1008 \times 1008$; patch size: $14 \times 14$). Most dropped patches belong to padding areas or uniform image backgrounds. All images from Wikimedia Commons.

OWL-ST L/14 self-trained on N-grams, not fine-tuned (Table 1 row 12)

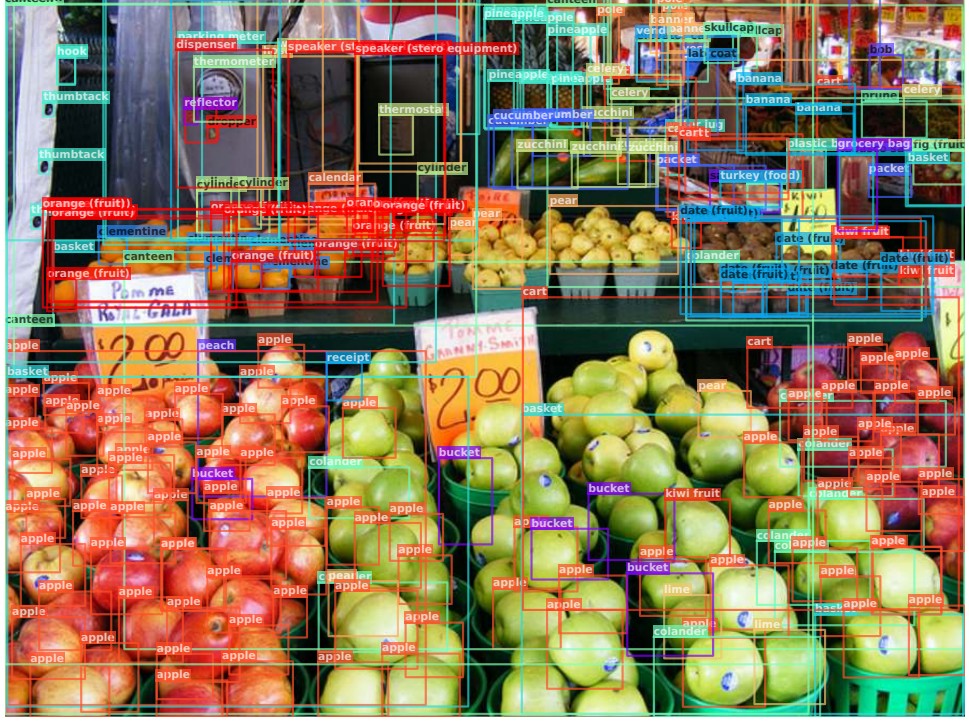

OWL-ST+FT L/14 self-trained on N-grams and fine-tuned on LVIS$_{base}$ (Table 1 row 15)

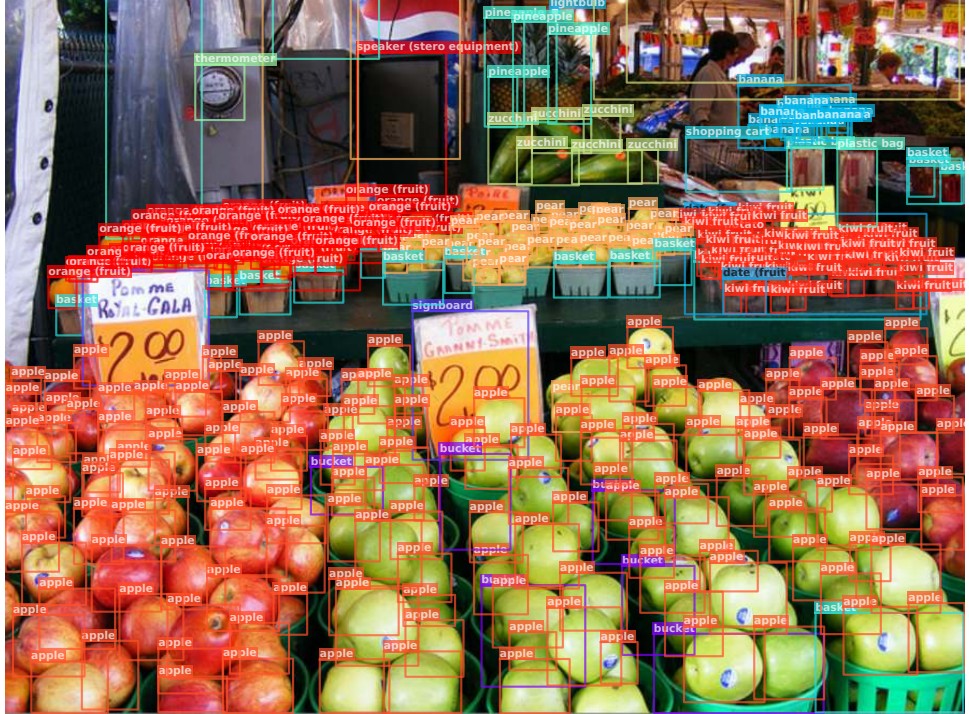

Figure A5: Qualitative example for OWLv2 L/14 from the LVIS val set. For the visualization, all LVIS classes were used as prompts. LVIS$_{rare}$ classes are labeled in black. **Top:** OWL-ST self-trained on N-grams, not fine-tuned (Table 1 row 12). **Bottom:** OWL-ST+FT self-trained on N-grams and fine-tuned on LVIS$_{base}$ (Table 1 row 15). Boxes above score 0.08 (top) or 0.3 (bottom) are shown.

OWL-ST L/14 self-trained on N-grams, not fine-tuned (Table 1 row 12)

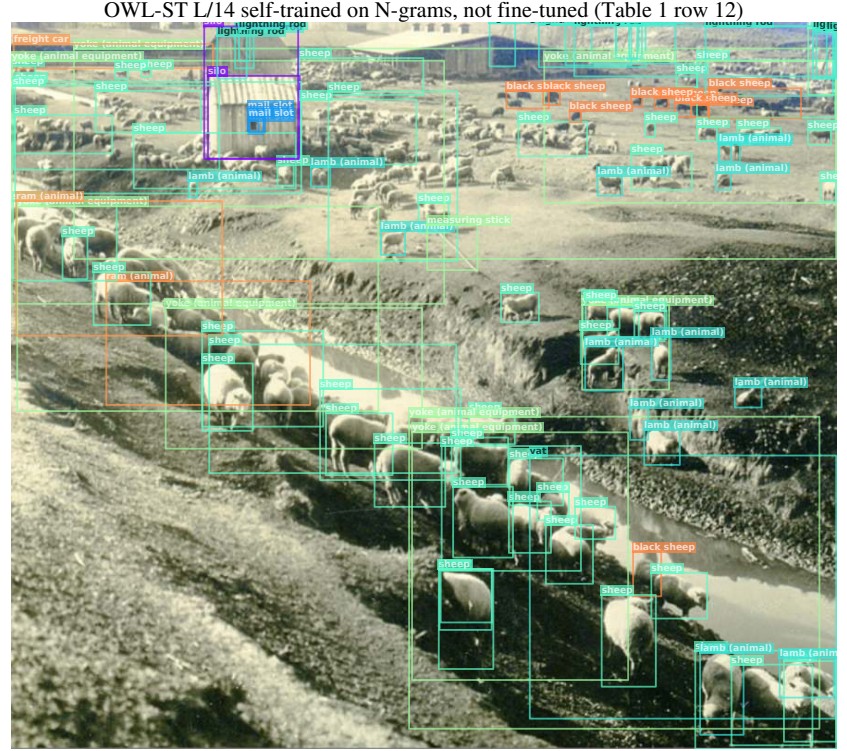

OWL-ST+FT L/14 self-trained on N-grams and fine-tuned on LVIS_base (Table 1 row 15)

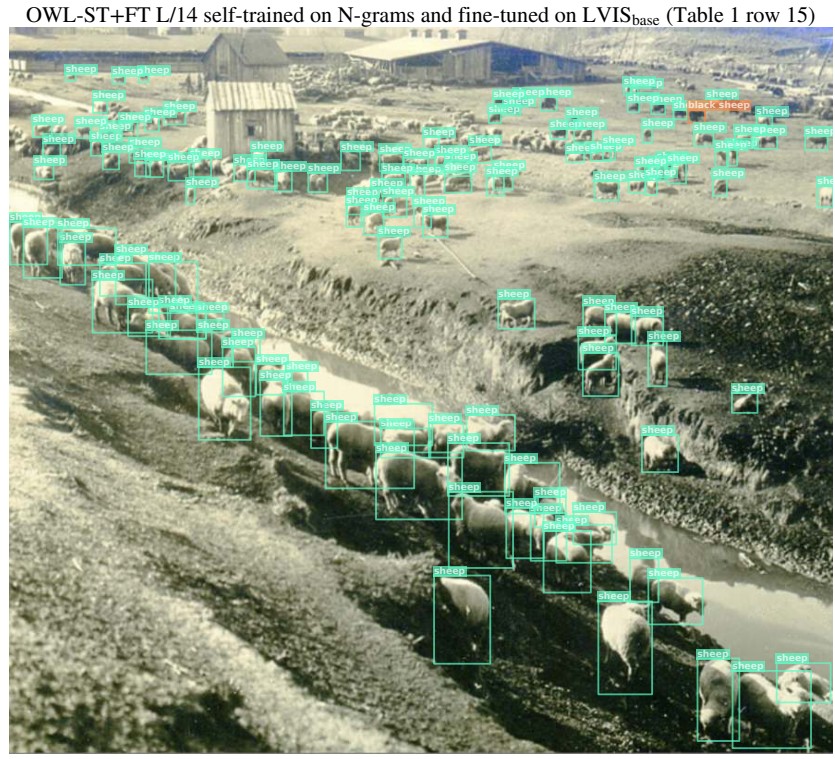

Figure A6: Qualitative example for OWLv2 L/14 from the LVIS val set. For the visualization, all LVIS classes were used as prompts. LVIS_rare classes are labeled in black. **Top:** OWL-ST self-trained on N-grams, not fine-tuned (Table 1 row 12). **Bottom:** OWL-ST+FT self-trained on N-grams and fine-tuned on LVIS_base (Table 1 row 15). Boxes above score 0.08 (top) or 0.3 (bottom) are shown.

OWL-ST L/14 self-trained on N-grams, not fine-tuned (Table 1 row 12)

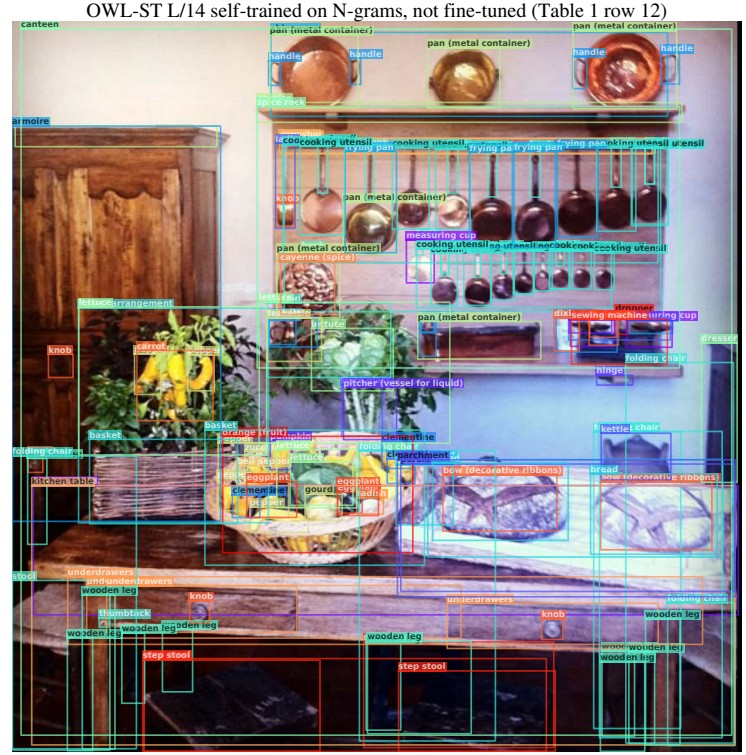

OWL-ST+FT L/14 self-trained on N-grams and fine-tuned on LVIS$_{base}$ (Table 1 row 15)

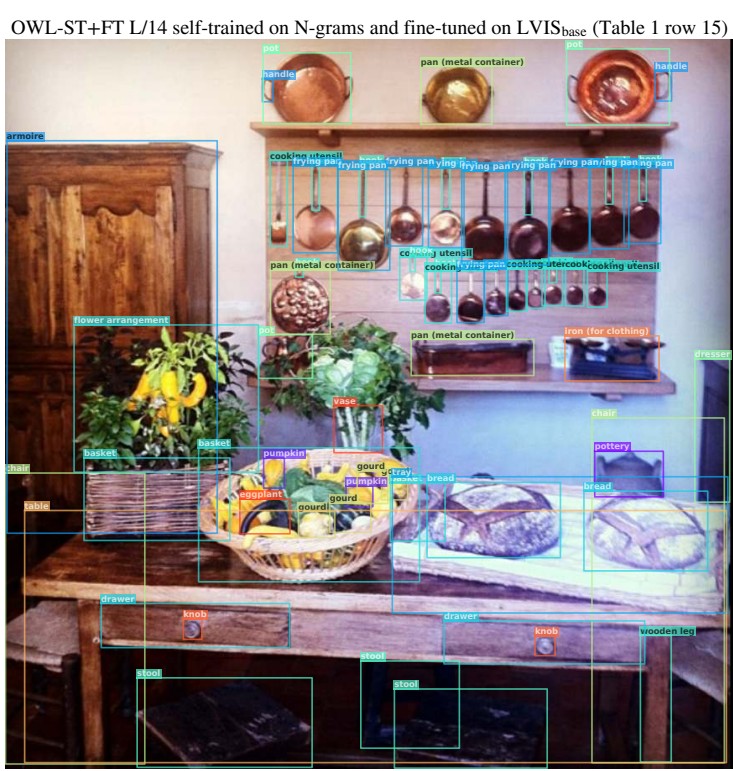

Figure A7: Qualitative example for OWLv2 L/14 from the LVIS val set. For the visualization, all LVIS classes were used as prompts. LVIS$_{rare}$ classes are labeled in black. **Top:** OWL-ST self-trained on N-grams, not fine-tuned (Table 1 row 12). **Bottom:** OWL-ST+FT self-trained on N-grams and fine-tuned on LVIS$_{base}$ (Table 1 row 15). Boxes above score 0.08 (top) or 0.3 (bottom) are shown.

