# OpenReview forum: "Scaling Open-Vocabulary Object Detection"
_NeurIPS.cc/2023/Conference — NeurIPS 2023 spotlight_

### Official Review · Reviewer_F7mV · 2023-06-12

**Soundness:** 3 good
**Presentation:** 3 good
**Contribution:** 3 good
**Rating:** 6
**Confidence:** 5

**Summary:**

The paper extends previous OWL-ViT detectors with self-training an 10B Web image-text data. The authors design different self-training strategies, such as token dropping, instance selection and Mosaics augmentation. It is an achievement to beat all previous methods on open-vocabulary benchmarks.

**Strengths:**

- The authors provide necessary implementation details about scaling up the open-vocabulary detector on massive web data with a self-training method.
- The performance on the open-vocabulary LVIS benchmark is truly impressive.
- The authors promise to open-source their code.

**Weaknesses:**

- GLIP2 performs better than OWL-ST on the ODinW dataset, can the authors explain the possible reasons?
- The authors may consider adding the ablation studies about w or w/o the mosaic augmentation. The additional gain from the mosaic augmentation is not clear.

**Questions:**

- What's the meaning of the $AP^{mini}$ metric in Table 1? Does it mean the LVIS test set?

**Limitations:**

I like this paper which includes comprehensive experiments and excellent performance, so I think this paper deserves to be accepted. But to me, this paper is not pretty interesting since this paper is more like a technical report with lots of engineering efforts rather than a research paper.

---

> ### Author Rebuttal · Authors · 2023-08-08
>
> Thank you for your review. Below, we address your questions and comments.
>
> ## Response to Weaknesses
> **_GLIP2 performs better than OWL-ST on the ODinW dataset, can the authors explain the possible reasons?_**
>
> GLIP2 uses much more human-annotated data than OWL-ST. Specifically, OWL-ST only uses Objects365 and Visual Genome (indirectly, through the annotator model). GLIP2 additionally trains on COCO, OpenImages, ImageNetBoxes, and GoldG (which includes 800'000 images with human grounding annotations from RefCOCO, RefCOCOg, and RefCOCO+). In addition, some of the tasks in ODinW are derived from OpenImages, so GLIP/GLIP2 is partially trained on ODinW evaluation data. We believe that these additional human annotations explain why GLIP reports higher numbers on ODinW, and will explore adding these to the OWL-ST training or fine-tuning data.
>
> **_The authors may consider adding the ablation studies about w or w/o the mosaic augmentation._**
>
> We performed an ablation study on the mosaic augmentation and provide it in the rebuttal PDF. We will add it to the paper appendix. The results show that increasing the number of mosaic tiles improves overall LVIS AP (top left plot) at least up to $12 \times 12$ (the maximum we tried). Improvements are primarily due to the "rare" classes and "small" objects. Note that the benefit of mosaics may be due to seeing smaller objects on average, or due to seeing more WebLI images per training step (a $12 \times 12$ mosaic contains 144 WebLI images).
>
> ## Response to Questions
> **_What's the meaning of the $AP^{mini}$ metric in Table 1? Does it mean the LVIS test set?_**
>
> $AP^{mini}$ refers to the LVIS "minival" set, which is a subset of the validation set and was introduced by MDETR (https://arxiv.org/pdf/2104.12763.pdf) and subsequently used by some object detection papers like GLIP/GLIPv2. We will clarify this in the table caption.
>
> We report $AP^{mini}$ for comparability with the papers that only report that metric. Note that $AP^{mini}$ values are significantly different (usually higher) from the standard LVIS AP.
>
> ## Response to Limitations
> **_...this paper is more like a technical report with lots of engineering efforts rather than a research paper._**
>
> Given the focus on scaling in the title and text of the paper, we understand this impression. However, we want to emphasize that **we do not simply scale up existing methods. On the contrary, we provide new research results which show that prior approaches to self-training for object detection were not optimal. These new insights improve performance independently of massive scale.** Specifically, through systematic study, we find that weak filtering and simple label space construction outperform prior approaches like strong filtering (e.g. using only a single pseudo-box or query per image) or complex grammatical query parsing. Figures 2, 3, and 5 all provide new research results that are applicable to other tasks, already at moderate data and compute scales (also see response to Reviewer 5UCC). We therefore believe that our work is not only a technical scaling effort but provides a valuable update to the research on training localization models.

---

### Official Review · Reviewer_tv8z · 2023-06-27

**Soundness:** 4 excellent
**Presentation:** 3 good
**Contribution:** 4 excellent
**Rating:** 7
**Confidence:** 4

**Summary:**

The paper proposes a self-training recipe for open-vocabulary object detection that leverages weak supervision in the form of image-text pairs from the Web. The authors identify three key ingredients for optimizing the use of weak supervision for detection: choice of label space, filtering of pseudo-annotations, and training efficiency. They propose to use all possible N-grams of the image-associated text as detection prompts for that image and apply only weak confidence filtering to the resulting pseudo-labels. The self-training recipe is applied to the OWL-ViT detection architecture and is called OWL-ST. The authors introduce OWLv2, an optimized architecture with improved training efficiency. The proposed method surpasses prior state-of-the-art methods already at moderate amounts of self-training and achieves further large improvements when scaled to billions of examples. The authors also evaluate their models on a suite of "in the wild" datasets and study the trade-off between fine-tuned and open-vocabulary performance.

**Strengths:**

This paper is well-written, easy to understand, and very rigorous.
This paper thoroughly discusses the methods and challenges of extending OWL-VIT to the billion-level image scale.
This paper achieves surprising performance.

**Weaknesses:**

As the authors mentioned, the biggest issue with this paper is the significant amount of computation required, which makes it difficult for general research institutions to reproduce. Therefore, my question is whether the authors will make the research results available as a foundational model, similar to SAM, for public use. Additionally, can it create a wave of downstream tasks, like SAM did, by providing basic capabilities?

**Questions:**

Since the authors have mentioned SAM, I am curious about how it compares to SAM. Although one is a segmentation model and the other is a detection model, SAM can also be combined with Grounded, such as in https://github.com/IDEA-Research/Grounded-Segment-Anything, to achieve detection. I'm wondering how the authors view the comparison between their model and SAM in terms of certain task capabilities. This is not a necessary question to answer, I just want to have a friendly discussion with the authors.

**Limitations:**

The main issue is still the enormous amount of computation required. However, if it is open-sourced as a foundational model for public use, it would still make a significant contribution to the community.

---

> ### Author Rebuttal · Authors · 2023-08-08
>
> Thank you for your review. Below, we address your questions and comments.
>
> ## Response to Weaknesses
> We are happy to confirm that we will make checkpoints and code publicly available. We hope that it will be useful as a foundation for downstream tasks.
>
> ## Response to Questions
> We see our work as complementary to SAM and think that it has large potential when combined with SAM, similar to the Grounded Segment Anything that you mentioned. SAM performs highly accurate conditioned segmentation, but it requires some spatial conditioning information (e.g. a box or a point) to know what to segment. Conditioning on text is only discussed briefly in the SAM paper and is not quantitatively evaluated. In contrast, we focus on the semantic aspect of object recognition, i.e. matching text descriptions to objects in an image. A combination of the two models, where OWLv2 is used to obtain boxes matching text queries, and SAM is then used to predict masks for the detected boxes, would provide a strong open-vocabulary segmentation solution.
>
> ## Response to Limitations
> Our fully trained models indeed required large amounts of compute. We will make these checkpoints publicly available to save other researchers the need for repeating the training. However, we would also like to point out that our method provides substantial improvements already at amounts of training that are comparable to previous models. In Figure 1 (right), the black cross indicates the amount of training used for the original OWL-ViT model. With similar training compute, and only about 250M additional web images, our L/14 model (orange line) improves over OWL-ViT by about 6 percentage points. We therefore make a useful contribution even at moderate amounts of compute.

---

> > ### Comment · Reviewer_tv8z · 2023-08-20
> >
> > After reading the other reviews and the authors' rebuttal, all of my questions have been well addressed. Therefore, I am inclined to maintain my rating.

---

### Official Review · Reviewer_EGtY · 2023-07-06

**Soundness:** 4 excellent
**Presentation:** 3 good
**Contribution:** 3 good
**Rating:** 7
**Confidence:** 4

**Summary:**

This paper presents OWLv2 model and OWL-ST self-training recipe for open-vocabulary object detection. The detection data is greatly enriched with the aid of self-training. Concretely, the authors use WebLI dataset as the source of weak supervision for self-training. The dataset consists of approximately 10B images and associated alt-text strings (noisy captions). Then OWL-ViT CLIP-L/14 is utilized to annotate all images with pseudo boxes. For self-training at scale, the authors present several techniques including Token dropping, Instance selection and Mosaics to make the training efficient. Experiments on LVIS show the effectiveness of the proposed approach.

**Strengths:**

1. This paper attemps to scale open-vocabulary object detection with self-training on large-scale weakly labeled dataset. This is a good engineering work built upon the existing detectors and datasets. This work verifies that large model + large data is a good option to enable zero-shot open-vocabulary detection.
2. The experimental results look like promising. Result on LVIS dataset is good.


**Weaknesses:**

1. Authors should carefully polish their paper and fixtypos such as: In L12, L/14->VIT/14; L109, consist->consists.
2. The authors are encouraged to discuss how to handle noise in self-training. What's effect without handling these noise.
3. What's the inference process? The authors are encouraged to describe this in their next version.

**Questions:**

See weaknesses box.

**Limitations:**

The limitations are adequately discussed in Section 5.

---

> ### Author Rebuttal · Authors · 2023-08-08
>
> Thank you for your review. Below, we address your questions and comments.
>
> ## Response to Weaknesses
> 1. **_Typos:_** Thank you for identifying these typos, we will fix them and do an additional round of proof-reading.
>
> 2. **_Discussion of noise in self-training:_** In the paper, we already discuss one way to handle noise in Section 4.4 (_Filtering of Pseudo-Annotations_). In this section, we filter pseudo-annotations by annotator confidence, which removes noisy annotations at the expense of biasing the remaining annotations to objects that the annotator is good at.
> We find that strong filtering (e.g. confidence threshold 0.7) tends to be worse than weak filtering (e.g. threshold 0.3), which suggests that bias introduced by strong filtering is worse than the noise in the data. Based on the trends in Figure 3 (i.e. confidence threshold 0.1 is worse than 0.3), we believe that no filtering (i.e. not handling noise at all) also performs poorly, suggesting that some filtering is necessary but that the remaining noise is acceptable. We will expand this discussion in the paper. Please let us know if you have other analyses in mind.
>
> 3. **_What's the inference process?_** The inference is the same as for the original OWL-ViT (see L187f: _"Note that at inference, the model is identical to the original OWL-ViT."_). We will expand this sentence and provide some details rather than only referring to the original OWL-ViT paper.

---

> > ### Comment · Reviewer_EGtY · 2023-08-20
> > **Thanks for your reply**
> >
> > My concerns have been addressed. I tend to keep my original rating.

---

### Official Review · Reviewer_5UCC · 2023-07-07

**Soundness:** 3 good
**Presentation:** 3 good
**Contribution:** 3 good
**Rating:** 7
**Confidence:** 4

**Summary:**

This work adopts a self-training strategy to generate web scale object annotations for open-vocabulary object detection. It uses an existing  open-vocabulary object detector (OWL-ViT) to annotate 10B image-text pairs, and uses the annotated dataset for self-training. A various architecture is also proposed to improve the training efficiency. Experimental results show a large performance gain on open-vocabulary detection.

**Strengths:**

1. A self-training strategy to annotate the bounding boxes for the large-scale image-text pairs. This provides an open-vocabulary detection dataset for various location-specific tasks, including detection, segmentation, visual grounding, etc. Despite self-training being widely explored in many previous works (discussed in Sec 2.3), this work provided a detailed ablation in the filtering strategy in Sec. 4.4.

2. The performance is promising in rare classes, indicating the ability of zero-shot transfer with the self-training strategy.

**Weaknesses:**

See questions.

**Questions:**

1. This paper is a coherent work of OWL-ViT. It's not straight forward for me to drop the DETR decoder in OWL, but rather predict a bounding box with each patch token. Such a dense prediction results in a significant number of background proposals (Instance selection in Sec. 3.2), which leads to inefficient detection.

2. While N-grams tend to maintain a higher number of image-text alignments in comparison to rule-based methods (such as highest-scoring or threshold-based approaches), they also tend to generate more redundant alignments, particularly for obvious objects where the confidence scores are higher (e.g., both 'boy', 'a boy', 'a boy in red' might correspond to the same region). Does this bias the model towards learning obvious objects while potentially overlooking the hard examples?

**Limitations:**

The self-training strategy in the paper requires a large amount of computing resources and web-crawled data, which is not feasible for many colleges and research institutes.

---

> ### Author Rebuttal · Authors · 2023-08-08
>
> Thank you for your review. Below, we address your questions and comments.
>
> ## Response to Questions
> 1. **_Is dropping the decoder inefficient?_** Our decoder-free architecture is actually more efficient than an equivalent encoder-decoder architecture like DETR. The number of encoder output tokens is the same in both cases. However, in a decoder model, every decoder layer cross-attends into the large number of encoder output tokens. Our encoder-only architecture only applies a lightweight objectness-prediction head to each encoder output token. Background tokens (= low objectness score) are then filtered out and do not need to be decoded into boxes (Instance Selection in Section 3.2). Therefore, the amount of processing that our model has to do per encoder output token is less than that of encoder-decoder models like DETR.
>
> 2. **_Do redundant ngrams bias the model towards learning obvious objects?_** No, because obvious objects do not contribute more pseudo-annotations than hard objects. Consider your example of an image containing a boy: OWL-ViT would predict one box for the boy in the image, and then assign high probabilities for the ngrams "boy", "a boy", "a boy in red" etc. There would be a large number of high-confidence, redundant ngrams for that box. However, for self-training, we only choose the single highest ngram for that box. This way, a box with many redundant matching ngrams will not have a higher weight in training than a box that has only a single matching ngram. Additionally, we use non-maximum suppression to remove near-duplicate pseudo-annotations. Further, we do not use "soft labels" during self-training, i.e. we treat low-confidence and high-confidence pseudo-boxes the same (as long as they meet the inclusion threshold). We will clarify this in the paper.
>
> ## Response to Limitations
> **_The self-training strategy in the paper requires a large amount of computing resources and web-crawled data, which is not feasible for many colleges and research institutes._** We would like to point out that our method provides substantial improvements already at amounts of training that are comparable to previous models and useful for resource-constrained researchers. In Figure 1 (right), the black cross indicates the amount of training used for the original OWL-ViT model. With similar training compute, and only about 250M additional web images, our L/14 model (orange line) improves over OWL-ViT by about 6 percentage points. Datasets with this amount of web data are publicly available (e.g. at https://laion.ai/). We therefore believe that this research is feasible with moderate resources and public data. Additionally, we will release self-trained model checkpoints.

---

> > ### Comment · Reviewer_5UCC · 2023-08-16
> > **Thanks for your reply**
> >
> > Most of my concerns are addressed. Despite this work being an extension of object detection self-training on a large dataset, it provides some insights into scaling up the detection dataset. Besides, the experimental results impressed me, thus my decision tends to accept.

---

### Author Rebuttal · Authors · 2023-08-08

We thank the reviewers for their useful feedback and comments.
The reviews were unanimously positive.
In our response, we address the remaining questions and concerns.
We provide an overview below and detailed responses to the individual reviewers.

In particular:

1. We confirm that we will release trained model checkpoints upon publication.
2. We now provide additional experiments to show the benefit of mosaics (see PDF; requested by Reviewer F7mV).
3. We clarify that we do not simply scale up existing methods, but provide new research results that challenge prior approaches to self-training and are useful independently of scale (see response to Reviewer F7mV).

We hope our work will be of value to the NeurIPS community both through the model release and through our research contributions.

---

### Decision · Program_Chairs · 2023-09-21

**Decision:**

Accept (spotlight)

**Comment:**

The paper scales up open-vocabulary object detection with improved computation efficiency and impressive results. The rebuttal provides additional ablation studies and successfully resolves the minor technical concerns. Five reviewers unanimously give accept recommendations. Clear accept. Scaling to large data and OOD scenarios is the direction of deep learning, so it is beneficial to have this paper seen by larger audience at the conference as a spotlight.